# Towards Effective Federated Graph Foundation Model via Mitigating Knowledge Entanglement

**Yinlin Zhu[1]\*, Xunkai Li[2]\*, Jishuo Jia[3], Miao Hu[1], Di Wu[1]†, Meikang Qiu[4]**

[1] Sun Yat-sen University, Guangzhou, China
[2] Beijing Institute of Technology, Beijing, China
[3] Shandong University, Weihai, China
[4] Augusta University, Augusta, Georgia, USA

`zhuylin27@mail2.sysu.edu.cn, cs.xunkai.li@gmail.com, jishuojia447@gmail.com`
`humiao5@mail.sysu.edu.cn, wudi27@mail.sysu.edu.cn, qiumeikang@yahoo.com`

## Abstract

Recent advances in graph machine learning have shifted to data-centric paradigms, driven by two emerging research fields: (1) Federated graph learning (FGL) facilitates multi-client collaboration but struggles with data and task heterogeneity, resulting in limited practicality; (2) Graph foundation model (GFM) enables desirable domain generalization but is typically confined to single-machine training, neglecting the potential of cross-silo data and computational resources. It is evident that these two paradigms are complementary, and their integration offers substantial advantages. Motivated by this, we present a pioneering study about the federated graph foundation model (FedGFM), a novel decentralized GFM training paradigm. Despite the promising vision of FedGFM, **_knowledge entanglement_** has emerged as a critical challenge, where multi-domain knowledge is encoded into indistinguishable representations, thereby limiting downstream adaptation.

To this end, we propose FedGFM+, an effective FedGFM framework with two key modules to mitigate knowledge entanglement in a dual-pronged manner. (1) **AncDAI**: From a global perspective, we introduce a novel **anc**hor-based **d**omain-**a**ware **i**nitialization strategy. Before pre-training, each client encodes its local graph into a domain-specific prototypes, which serve as semantic anchors in the representation space. Around each anchor, we construct synthetic embeddings to initialize the global model. We theoretically show that these prototypes are distinguishable across domains, and the initialization provides a strong inductive bias that facilitates disentanglement of domain-specific knowledge. (2) **AdaDPP**: From a local perspective, during pre-training, each client independently learns a lightweight graph prompt that captures domain semantic preferences. During fine-tuning, prompts from all clients are aggregated into an **ada**ptive **d**omain-sensitive **p**rompt **p**ool, from which the GFM selects relevant prompts to augment the target graphs attributes, thereby improving the downstream adaptation. FedGFM+ is extensively evaluated on 8 diverse benchmarks spanning multiple domains and tasks, outperforming 20 baselines from isolated supervised learning, FGL, and federated variants of centralized GFM paradigms.

---

\*Equal contribution.
†Corresponding author.

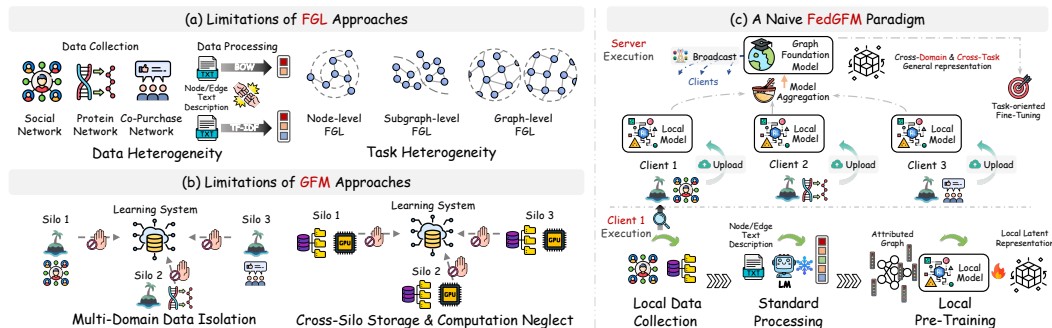

Figure 1: Comparison of the FGL, GFM, and naive FedGFM paradigm. (a) Limitations of FGL approaches; (b) Limitations of GFM approaches; (c) A naive FedGFM paradigm organically combines the complementary strengths of FGL and GFM to overcome their respective limitations.

# 1 Introduction

Recent advances in computational capabilities have sparked a data-centric paradigm shift in deep learning. Moving beyond an exclusive reliance on architectural innovations, the AI community now prioritizes large-scale data utilization, as evidenced by the success of GPT-4 [1] in language processing and Sora [30] in vision tasks. This data-centric scaling trend also extends to graph machine learning, where two learning paradigms are gaining prominence (1) Federated graph learning (FGL) enables cross-silo graph collaboration; (2) Graph foundation models (GFM) promote multi-domain graph generalization. However, both of them face practical deployment limitations.

Two limitations hinder FGL from achieving cross-domain and cross-task collaboration, as illustrated in Fig. 1 (a): (1) **Data Heterogeneity.** Due to diverse data sources and processing methods, client graphs often differ in feature dimension, label space, and topology pattern. As a result, most FGL methods are confined to collaboration across subsets of a single dataset [62, 26, 20]. While GCFL+ [53] and FedStar [36] enable limited cross-domain collaboration via domain-aware client clustering or feature-agnostic parameter sharing, they are only applicable to graph-level tasks and lack the ability to capture cross-domain general knowledge at the feature level. (2) **Task Heterogeneity.** Existing FGL assumes uniform graph granularity and downstream tasks across clients, enforcing one of three settings: node-level (ego-networks for node classification/link prediction), subgraph-level (induced subgraphs from a global graph for node classification/link prediction), or graph-level (graph sets for classification/regression) [16]. As a result, existing FGL approaches often adopt task-specific designs in both model architectures and training algorithms, which significantly limits their ability to support collaboration across multi-task graph data.

Meanwhile, existing GFM studies face the following two limitations, as illustrated in Fig. 1 (b): (1) **Multi-Domain Data Isolation.** Training generalizable GFMs requires diverse graph data spanning multiple domains, like social networks, molecular structures, etc. Although a number of public graph datasets are available, they remain limited in both scale and diversity. In contrast, real-world graph data is expected to continuously grow in volume and variety, yet it is often distributed across institutions and isolated in data silos due to privacy regulations or commercial competition. This renders existing centralized GFM approaches increasingly infeasible. (2) **Cross-Silo Storage and Computation Neglect.** Although current GFMs require significantly fewer storage and computation resources than their NLP or vision counterparts, which makes them feasible within a single institution, centralized training frameworks inherently fail to leverage the vast yet fragmented storage and computation capacities distributed across multiple silos in real-world deployments. This under-utilization results in non-trivial opportunity costs, such as redundant resource provisioning and sub-optimal training efficiency.

Fortunately, FGL and GFM exhibit a naturally complementary relationship. Specifically, FGL equips GFM with a decentralized training paradigm that supports learning across distributed silos while efficiently utilizing cross-silo storage and computational resources. In contrast, GFM enhances FGL by offering unified feature encoding and a pre-training followed by fine-tuning framework, thereby facilitating generalized collaboration across diverse graph domains and task types. To this end, we introduce **Federated Graph Foundation Model** (FedGFM), a novel and practical paradigm designed for training GFM over decentralized, cross-domain, and cross-task graphs. As illustrated in

Fig. 1 (c), the FedGFM paradigm follows a pipeline that begins with federated pre-training and proceeds with fine-tuning. During the federated pre-training phase, each client performs self-supervised learning on its private graph to acquire domain-specific representations. The server then aggregates these local models to construct a global model that captures generalizable topological and semantic patterns. The global model is subsequently broadcast to clients as the initialization for the next round of federated pre-training. This iterative process continues across multiple rounds of federated communication. In the fine-tuning phase, the global model is treated as a graph foundation model and is further adapted to specific downstream tasks through supervised learning.

To establish an effective FedGFM framework, our work begins with an empirical investigation (Sec. 3), assessing its feasibility and revealing a non-trivial challenge. Specifically, (1) From a feasibility perspective, FedGFM faces stringent communication constraints, as frequent transmission of large-scale model parameters or gradients is often impractical in real-world federated deployments. This limitation calls for a lightweight yet expressive model architecture. Fortunately, the graph vector quantization-variational auto-encoder (gVQ-VAE), which is widely used as the backbone in centralized GFM, presents a promising solution. It has been extensively validated for its ability to jointly encode graph structures and text attributes into discrete, semantically meaningful representations [41, 43], making it well-suited for multi-domain pre-training. Meanwhile, its lightweight design naturally aligns with the communication-efficiency requirements of FedGFM. (2) However, naively distributing the pre-training of gVQ-VAE across local clients in a federated setting introduces a critical challenge we term knowledge entanglement. Unlike centralized training, federated pre-training operates on multiple isolated, domain-specific graphs, each with distinct data distributions. Each client's local trained model tend to overfit their domain-specific data without alignment across clients. Consequently, the aggregated global GFM encodes multi-domain graphs into indistinguishable representations and further limits its downstream generalization.

Building upon these insights, we present an effective FedGFM framework named FedGFM+, which involves two key modules to mitigate knowledge entanglement in a dual-pronged manner: (1) **AncDAI**: From a global perspective, we introduce a novel **anc**hor-based **d**omain-**a**ware **i**nitialization strategy. Before pre-training, each client encodes its local graph into a domain-specific prototype, which serve as semantic anchors in the representation space. Around each anchor, we construct synthetic embeddings to initialize the global model. We theoretically show that these domain prototypes are distinguishable across domains, and the initialization provides a strong inductive bias that naturally facilitates encourages separation among knowledge representations from different domains. (2) **AdaDPP**: From a local perspective, during the pre-training stage, each client independently learns and retains a lightweight, domain-sensitive prompt that captures its local semantic preferences, without participating in federated aggregation. In the fine-tuning stage, these prompts are assembled into an **ada**ptive **d**omain-sensitive **p**rompt **p**ool. For a given target graph, the model selects and incorporates the most relevant prompts from the pool based on its semantic characteristics. These prompts serve as domain-specific priors that condition the GFMs representations, thereby enabling adaptive exploitation of domain knowledge and facilitating improved adaption to downstream tasks.

**Our Contributions.** (1) **Problem Identification.** To the best of our knowledge, this is the first exploration of the FedGFM paradigm, which organically combines FGL and GFM to offer a practical solution for training graph foundation model across silos with diverse graph domain and tasks. (2) **In-depth Investigation.** (Sec. 3) We conduct an in-depth empirical investigation for FedGFM, assessing its feasibility and revealing a non-trivial challenges named knowledge entanglement, providing valuable insights for its development. (3) **Novel Framework.** (Sec. 4) We propose a novel and effective FedGFM framework named FedGFM+, which employs two key modules to address the knowledge entanglement challenge, including AncDAI from the global perspective and AdaDPP from the local perspective. (4) **State-of-the-art Performance.** (Sec. 5) Extensive experimental results on graph learning with 8 cross-task and cross-domain datasets demonstrate the superiority of FedGFM+ compared with 20 baselines, including 5 isolated supervised learning methods, 10 FGL techniques, and 5 federated variants of centralized GFM training strategies.

## 2 Preliminaries and Problem Formalization

**Text-Attributed Graph.** Consider a text-attributed graph $G = (\mathcal{V}, \mathcal{E})$, where $\mathcal{V}$ is the set of nodes and $\mathcal{E}$ is the set of edges. Each node $v_i \in \mathcal{V}$ and edge $e_i \in \mathcal{E}$ may be associated with a textual description, which is encoded into a semantic vector using a specific embedding technique (e.g.,

bag-of-words, pre-trained language models). Depending on the downstream task, the graph may be equipped with supervision signals at different levels: node-level labels (for node classification), edge-level labels (for edge classification or link prediction), or graph-level labels (for graph classification).

**Graph Vector Quantization-Variational Auto-Encoder as GFM Backbone.** Most recent GFMs adopt gVQ-VAEs as the trainable GNN. This backbone enables the joint encoding of topology and textual attributes into a discrete embedding space with clear semantic boundaries, making it particularly suitable for multi-domain GFM pre-training. Specifically, (1) $\mathcal{G}' = (\mathcal{V}, \mathcal{E}, \mathcal{X}) \rightarrow \textbf{\textit{Encoder}} \rightarrow$ *Embeddings*: To ensure generality in arbitrary inputs, the Encoder can be instantiated as any reasonable GNN capable of incorporating both node and edge features to generate informative embeddings $z \in \mathbb{R}^d$. (2) *Embeddings* $\rightarrow \textbf{\textit{Codebook}} \rightarrow$ *Quan. Emb.*: To establish clear semantic boundaries, the Codebook $\mathcal{C}$ transforms continuous embeddings $z$ into discrete embeddings $e \in \mathbb{R}^d$ (Quan. Emb. $z_q \in \mathbb{R}^d$) via similarity retrieval-based vector quantization:

$$z_q \leftarrow e_j, \ j = \arg\min_{e_i \in \mathcal{C}} \|z - e_i\|_2, \ \mathcal{C} = \{e_1, e_2, \ldots, e_K\}. \tag{1}$$

(3) *Quan. Emb.* $\rightarrow \textbf{\textit{Decoder}} \rightarrow \mathcal{G}'_r = (\mathcal{V}, \mathcal{E}_r, \mathcal{X}_r)$: To enable the self-supervised training, gVQ-VAEs follow an autoencoder framework, where gradients are computed by the discrepancy between the reconstructed graph $\mathcal{G}'_r$ and the original input graph $\mathcal{G}'$, thereby updating the Encoder and Codebook. Notably, the trainable components of the Encoder and the Codebook are the weighted matrix and the discrete embeddings $\{e_1, \ldots, e_K\}$, which together constitute the trainable GFM embedding function parameterized by $f_\theta$. Meanwhile, to construct end-to-end gradient flow, the straight-through estimator (STE) [4] is used to approximate gradients by bypassing the non-differentiable quantization step. Formally, the gVQ-VAE is pre-trained via optimizing loss function as follows:

$$\mathcal{L}_{pretrain} = \mathcal{L}_{feat} + \mathcal{L}_{topo} + \frac{1}{n}\sum_{i=1}^{n}\big\|\text{sg}[z_i] - z_{q_i}\big\|_2^2 + \cdot\frac{1}{n}\sum_{i=1}^{n}\big\|z_i - \text{sg}[z_{q_i}]\big\|_2^2,$$

$$\mathcal{L}_{feat} = \frac{1}{n}\sum_{i=1}^{n}(1 - \frac{x_i^T \hat{x}_i}{||x_i|| \cdot ||\hat{x}_i||})^\gamma, \qquad \mathcal{L}_{topo} = ||A - \sigma(\hat{X}\hat{X}^T)||_2^2, \tag{2}$$

where $\text{sg}[\cdot]$ represents the stop-gradient operator, $n$ denotes the number of nodes, $z_i$ represents the $i$-th node embedding produced by the GNN encoder, $z_{q_i}$ denotes its quantized embedding obtained by retrieving the codebook, and $\hat{x}_i$ denotes the reconstructed node attributes projected via MLP-based decoders, i.e., $\hat{x}_i = \delta(z_{q_i})$, $\gamma$ is the scaling factor. More details and related works about gVQ-VAE are presented in Appendix A.

**Problem Formalization of FedGFM.** For FedGFM, there is a trusted central server and $K$ clients. The subgraphs or graph collections of the client present a relationship such as subgraph-level decentralization or graph-level decentralization (see Appendix. C.2 for more details about data settings). To unify the representation, we regard the graph data held by $k$-th client as $\mathcal{S}_k$, where $|\mathcal{S}_k| = 1$ for subgraph-level decentralization. The proposed FedGFM paradigm follows a federated pre-training-fine-tuning process. For the **Federated Pre-Training** phase, each client conducts self-supervised training to optimize its local model based on its local graph, and the server aggregates multiple local models to obtain a global graph foundation model. Consider adapting the widely-used FedAvg [32] aggregation strategy in federated learning for vision tasks within the FedGFM framework, the federated pre-training process unfolds as follows: (1) Initialization: At the first communication round ($r=1$), the central server sets the local model parameters of $K$ clients to the global parameters, i.e., $\Theta^k \leftarrow \Theta^g \ \forall k$. (2) Local Updates: Each local model performs training on the current local data $G^k$ to minimize the self-supervised loss $\mathcal{L}(G^k; \Theta^k)$, and then updating the parameters: $\Theta^k \leftarrow \Theta^k - \eta\nabla\mathcal{L}$. (3) Global Aggregation: After local training, the server aggregates the local knowledge with respect to the number of training instances, i.e., $\Theta^g \leftarrow \frac{N_k}{N}\sum_{k=1}^{K}\Theta^k$ with $N = \sum_k N_k$, and distributes the global parameters $\Theta^g$ to local clients selected at the next round. This process iterates between steps 2 and 3 until reaching the final round $R$. This iterative cycle continues until the completion of the last round ($r=R$), facilitating collaborative GFM training by parameter sharing without the exchange of private local data. For the **Fine-Tuning** phase, FedGFM first loads and freezes the pre-trained global model from the central server as GFM, then uses available graph supervision signals to fine-tune the task heads to adapt to specific downstream graph tasks.

# 3 Empirical Investigation

In this section, we present an in-depth empirical study of the FedGFM paradigm, organized around two key questions from different perspectives. **Q1**: From the perspective of **Feasibility**, is FedGFM practical for real-world deployment? **Q2**: From the perspective of **Effectiveness**, what are the main bottlenecks that limit the effectiveness of a naive FedGFM implementation?

Table 1: Comparison of parameter sizes between graph foundation models and those in the language and vision fields. Parameter counts are shown above each method name. '*' indicates an upper bound. Graph, Language and vision models are highlighted in red, yellow and blue, respectively.

| 16.8M | 20M | 20M | 40M |
| --- | --- | --- | --- |
| AnyGraph [49] | GFSE [9] | SwapGT [10] | OpenGraph [51] |
| 25M | $10M^*$ | $5M^*$ | 180M |
| OFA [29] | GOFA [23] | GQT [41] | Unigraph [17] |
| 7M | 10M | 150M | $31.64M^*$ |
| GFT [43] | RAGraph [21] | GraphCLIP [67] | UniGraph2 [18] |
| 7B | 175B | $540B^*$ | 1B |
| Llama2-7B [38] | GPT-3 [6] | PaLM [11] | DINOv2 [67] |

To address **Q1**, we survey several representative foundation models to quantify their parameter scales, and summarize the results in Table 1. Notably, compared with foundation models in language and vision domains, graph foundation models (GFMs) are significantly more lightweight in terms of parameter size. This suggests that federated pre-training of GFMs is communication-efficient and practically feasible. Among all surveyed GFMs, we further observe that two gVQ-VAE-based methods, GFT [43] and GQT [41], exhibit the smallest parameter scales. This highlights the advantage of the gVQ-VAE architecture in achieving a lightweight yet expressive design, making it particularly suitable for FedGFM settings. More related works about GFM are presented in Appendix A.

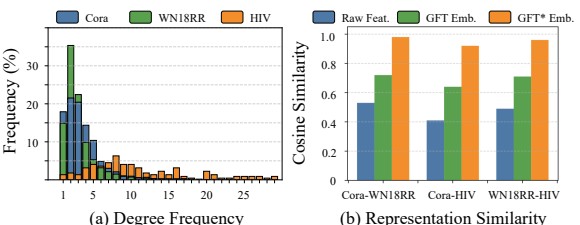

Figure 2: Empirical analysis on three graph datasets: Cora, WN18RR, and HIV. (a) Comparison of topological patterns in terms of degree distribution. (b) Average cosine similarity of original node features and node embeddings encoded by GFT and $GFT^*$, respectively.

To address **Q2**, we conduct a simple yet illustrative visualization experiment, aiming to reveal the bottlenecks that limit the effectiveness of naive FedGFM. Building on the insight of **Q1**, we implement naive federated variants of GFT [43] (denoted as $GFT^*$), and evaluate GFT and $GFT^*$ on three datasets: Cora [56], WN18RR [12], and HIV [47], covering different domains (citation networks, knowledge graphs, and molecular graphs).

The empirical results are presented in Fig. 2. Specifically, panel (a) illustrates the node degree distributions of the Cora, WN18RR, and HIV datasets (restricted to the first 30 degrees starting from 1 for visual clarity), while panel (b) reports the inter-domain cosine similarity among the three datasets, computed in three different representation spaces: (1) the average initial node features, (2) the average node embeddings learned by GFT, and (3) those learned by $GFT^*$. This comparison reveals how well each model distinguishes multi-domain knowledge during representation learning. As observed, the three datasets differ markedly in both topological structure and initial feature distributions. Despite such heterogeneity, centralized GFT pretraining produces a graph foundation model that generates embeddings with clear domain-specific distinctions. This indicates effective preservation of inter-domain variability through joint optimization. In contrast, the embeddings learned by $GFT^*$ under decentralized federated pretraining show near-unity inter-domain similarity, reflecting a collapse of domain specificity caused by the absence of coordinated global optimization. We term this the **knowledge entanglement**, a non-trivial challenge to resolve for effective FedGFM design.

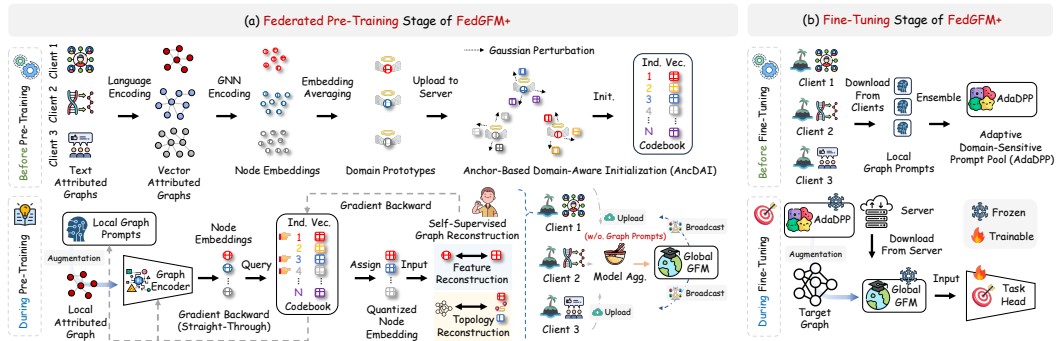

Figure 3: Overview of the proposed FedGFM+ framework.

# 4 Methods

In this section, we introduce the proposed FedGFM+ framework. We first provide an overview of FedGFM+ in Fig. 3. At its core, FedGFM+ adopts a federated pre-training and fine-tuning paradigm. During each communication round of pre-training, clients leverage a local gVQ-VAE encoder to perform self-supervised graph reconstruction, capturing domain-specific semantics. The resulting local models are uploaded to the server for aggregation, yielding an updated global model. The global model is subsequently broadcast to clients as the initialization for the next round of federated pre-training. In the fine-tuning stage, this global model serves as a general-purpose GFM encoder, while a task-specific prediction head is optimized for downstream tasks. Moreover, FedGFM+ introduces two key modules to mitigate the knowledge entanglement challenges: (1) **AncDAI**: Before pre-training, FedGFM+ employs a novel anchor-based domain-aware initialization strategy to initialize the global codebook, providing a strong inductive bias that facilitates disentanglement of domain-specific knowledge. (2) **AdaDPP**: During pre-training, each client independently learns a lightweight graph prompt that imbues the GFM with its own domain semantic preferences. During fine-tuning, prompts from all clients are aggregated into an adaptive domain-sensitive prompt pool, from which the GFM selects relevant prompts to augment the target graph attributes, thereby improving the downstream adaptation. Below we introduce these two modules in detail.

## 4.1 Anchor-Based Domain-Aware Initialization

As discussed in Section 3, naive FedGFM suffers from knowledge entanglement, where representations from different domains collapse into indistinguishable embeddings. To mitigate this, from a global perspective, we aim to endow the global model with a strong inductive bias that explicitly encourages the separation of domain-specific semantics.

Before federated pre-training, to capture domain-specific knowledge, we introduce a domain prototype extraction mechanism, which models intrinsic patterns in the graph topology and node attributes of the local graph and summarizes them into a compact, unified-dimensional vector representation. Specifically, for the $k$-th client with a local graph $\mathcal{G}^k = (\mathcal{V}^k, \mathcal{E}^k)$, node features $\mathbf{X}^k$ and adjacency matrix $\mathbf{A}^k$, we first compute the node embeddings $\mathbf{Z}^k$ as follows:

$$\mathbf{Z}^k = f_{\theta^{\mathrm{glb}}}(\mathbf{X}^k, \mathbf{A}^k) \tag{3}$$

where $\theta^{\mathrm{glb}}$ denotes the initialized global model parameter broadcast to all clients. The domain prototype $\mathbf{p}^k$ is then obtained by mean-pooling over node embeddings:

$$\mathbf{p}^k = \frac{1}{|\mathcal{V}^k|} \sum_{i \in \mathcal{V}^k} \mathbf{z}_i^k \tag{4}$$

We theoretically demonstrate that, even under a randomly initialized model with shared parameters, the domain prototypesobtained via averaging the encoded node representationsremain distinguishable across clients. This separability stems from intrinsic discrepancies in node features and graph topologies among domains, and can be formally bounded (Appendix B Theorem. B.1).

Each client subsequently uploads its prototype to the central server. To steer the global model toward learning domain-aware representations, we treat these prototypes as semantic anchors and

synthesize local neighborhoods in the embedding space via controlled perturbations. Specifically, for each anchor $\mathbf{p}^k$, a set of perturbed embeddings $\{\tilde{\mathbf{p}}_i^k\}_{i=1}^H$ is generated as:

$$\tilde{\mathbf{p}}_i^k = \mathbf{p}^k + \sigma\epsilon_i, \quad \epsilon_i \sim \mathcal{N}(\mathbf{0}, \mathbf{1}), \quad i = 1, \ldots, H, \tag{5}$$

where $\epsilon_i$ is sampled from a standard Gaussian distribution, and $\sigma$ is a noise scaling factor that ensures numerical stability. Notably, the number of synthetic embeddings $H$ is uniformly allocated across prototypes, depending on the number of the learnable codebook tokens in the global model.

Finally, the synthetic embeddings aggregated from all domains are used to initialize the codebook $\mathcal{C}$ of the global model, i.e., $\mathcal{C} \leftarrow \text{Init}(\cup_k\{\tilde{\mathbf{p}}_i^k\}_{i=1}^H)$. We further provide a theoretical analysis (Appendix B Theorem. B.2) to demonstrate that this initialization introduces a structured inductive bias, which not only facilitates disentangled representation learning across diverse domains but also stabilizes optimization during the early stages of federated pretraining.

## 4.2 Adaptive Domain-Sensitive Prompt Pool

Moreover, to address knowledge entanglement from the local perspective, we introduce a novel prompt learning-based mechanism. During the pre-training stage, each client independently learns and retains domain-specific prompts and is excluded from federated aggregation. During the fine-tuning stage, these prompts serve as semantic priors that condition the GFM's representations, facilitating improved adaptation to diverse downstream tasks.

Concretely, during federated pre-training, each client maintains a set of learnable prompt tokens embedded in its local graphs feature space. For the $k$-th client, this prompt set is denoted as $\Phi^k = \{\phi_i^k\}_{i=1}^\lambda$, where $\lambda$ is the number of prompts and $F$ the feature dimensionality. Given the local graph $G^k = (\mathcal{V}^k, \mathcal{E}^k)$ and node features $\{x_i^k\}_{v_i \in \mathcal{V}^k}$, node representations are enhanced by a weighted combination of prompts, with attention weights computed via $\lambda$ learnable linear projections:

$$\tilde{x}_i^k = x_i^k + \sum_{j=1}^\lambda \alpha_j^k \phi_j^k, \quad \alpha_j^k = \frac{e^{(\mathbf{w}j^k)^T x_i^k}}{\sum_{t=1}^\lambda e^{(\mathbf{w}_t^k)^T x_i^k}}, \tag{6}$$

where $\alpha_j^k$ reflects the relevance of the $j$-th prompt to node $v_i$, and $\mathbf{w}_j^k$ is the corresponding learnable projection vector. These prompts and projection weights are optimized together with the local GNN backbone through a self-supervised graph reconstruction task, as described in Eq. 2.

During the fine-tuning stage, we downloads the global model as GFM, which encodes generalizable cross-domain knowledge. In parallel, it collects all locally learned prompts and associated projection weights to construct a adaptive domain-aware prompt pool, denoted as $\rho = \{\phi_i^j\}_{i=1,j=1}^{\lambda,K}$ and $\mathbf{w} = [\mathbf{w}^1, \ldots, \mathbf{w}^K]$. Given a target graph $G^{\text{tgt}} = (\mathcal{V}^{\text{tgt}}, \mathcal{E}^{\text{tgt}})$, node features are augmented using this prompt pool. For each node $v_i \in \mathcal{V}^{\text{tgt}}$ with feature $x_i^{\text{tgt}}$, the enhanced representation is computed as:

$$\tilde{x}_i^{\text{tgt}} = x_i^{\text{tgt}} + \sum_{p=1}^K \sum_{j=1}^\lambda \alpha_j^p \phi_j^p, \quad \alpha_j^p = \frac{e^{(\mathbf{w}j^p)^T x_i^{\text{tgt}}}}{\sum_{t=1}^K \sum_{l=1}^\lambda e^{(\mathbf{w}_l^t)^T x_i^{\text{tgt}}}}. \tag{7}$$

As a result, FedGFM+ effectively capitalizes on domain-specific prompts acquired during pre-training, substantially improving its adaptability to heterogeneous domains and diverse downstream tasks in the fine-tuning phase.

## 5 Experiments

In this section, we present a comprehensive evaluation of FedGFM+. We begin by introducing the experimental setup (Sec.5.1), and then seek to answer the following research questions: **Q1**: After task-specific fine-tuning, does the GFM trained by FedGFM+ consistently outperform (1) isolated supervised learning techniques, (2) state-of-the-art FGL baselines, and (3) naive federated variants of centralized GFM strategies across node-, edge-, and graph-level prediction tasks (Sec.5.2)? **Q2**: How does each individual module contribute to the overall performance of FedGFM+ (Sec.5.3)? **Q3**: Is FedGFM+ robust to changes in hyperparameter configurations (Sec.5.4)? In addition to the main evaluation, we further investigate the few-shot generalization ability (**Q4**) in Appendix D.

Table 2: Performance comparison of FedGFM+ and baselines. Best results of each baseline category are in underline. '*' denotes federated variants of centralized GFM. 'N/A' denotes task inapplicability. Node, edge, and graph classification datasets are marked in red, yellow, and blue, respectively.

| Method \ Dataset | Cora | PubMed | OGB-arxiv | WikiCS | FB15K237 | WN18RR | HIV | PCBA |
|---|---|---|---|---|---|---|---|---|
| Linear | 73.44 ± 0.13 | 85.11 ± 0.15 | 67.55 ± 0.08 | 74.38 ± 0.16 | 72.05 ± 0.14 | 84.33 ± 0.20 | 65.48 ± 0.23 | 57.71 ± 0.22 |
| GCN [22] | 80.17 ± 0.35 | 84.70 ± 0.22 | 72.50 ± 0.24 | 77.24 ± 0.16 | 71.24 ± 0.30 | 82.27 ± 0.18 | 65.37 ± 0.51 | 63.41 ± 0.20 |
| GAT [39] | 81.09 ± 0.33 | 84.47 ± 0.11 | 71.34 ± 0.29 | 77.59 ± 0.19 | 73.07 ± 0.19 | 85.52 ± 0.12 | 65.02 ± 0.28 | 64.83 ± 0.26 |
| GraphSAGE [15] | 80.52 ± 0.28 | 85.20 ± 0.24 | 72.78 ± 0.31 | 77.63 ± 0.21 | 72.10 ± 0.38 | 82.98 ± 0.22 | 65.19 ± 0.27 | 66.42 ± 0.14 |
| GIN [54] | 78.45 ± 0.23 | 83.61 ± 0.44 | 70.74 ± 0.37 | 69.24 ± 0.25 | 70.06 ± 0.14 | 80.25 ± 0.28 | 66.30 ± 0.18 | 68.83 ± 0.30 |
| FedAvg [32] | 81.45 ± 0.27 | 85.22 ± 0.18 | 71.53 ± 0.29 | 77.67 ± 0.13 | 73.14 ± 0.11 | 83.55 ± 0.20 | 66.05 ± 0.15 | 68.52 ± 0.28 |
| MOON [24] | 81.72 ± 0.38 | 85.84 ± 0.21 | 72.50 ± 0.41 | 77.54 ± 0.24 | 73.20 ± 0.15 | 83.64 ± 0.45 | 67.10 ± 0.26 | 69.81 ± 0.30 |
| FedSage+ [62] | 82.15 ± 0.28 | 86.37 ± 0.15 | 72.80 ± 0.11 | 78.64 ± 0.34 | 73.17 ± 0.22 | 82.95 ± 0.16 | N/A | N/A |
| Fed-PUB [3] | 81.98 ± 0.20 | 86.51 ± 0.32 | 73.15 ± 0.29 | 78.32 ± 0.43 | 72.84 ± 0.13 | 83.79 ± 0.25 | N/A | N/A |
| FedGTA [26] | 82.41 ± 0.33 | 87.10 ± 0.25 | 73.28 ± 0.14 | 78.60 ± 0.24 | N/A | N/A | N/A | N/A |
| FedTAD [66] | 82.24 ± 0.18 | 86.95 ± 0.30 | 72.50 ± 0.17 | 78.22 ± 0.27 | N/A | N/A | N/A | N/A |
| FGSSL [20] | 81.55 ± 0.42 | 85.60 ± 0.21 | 73.33 ± 0.15 | 76.25 ± 0.24 | N/A | N/A | N/A | N/A |
| FGGP [40] | 82.03 ± 0.13 | 85.10 ± 0.37 | 74.19 ± 0.05 | 76.44 ± 0.18 | N/A | N/A | N/A | N/A |
| GCFL+ [53] | N/A | N/A | N/A | N/A | N/A | N/A | 67.51 ± 0.14 | 71.95 ± 0.28 |
| FedStar [36] | N/A | N/A | N/A | N/A | N/A | N/A | 67.82 ± 0.21 | 71.27 ± 0.39 |
| OFA* [29] | 80.04 ± 0.33 | 85.30 ± 0.29 | 73.12 ± 0.25 | 78.55 ± 0.37 | 72.88 ± 0.26 | 84.28 ± 0.49 | 67.00 ± 0.19 | 71.05 ± 0.28 |
| GFT* [43] | 81.07 ± 0.24 | 84.24 ± 0.38 | 73.19 ± 0.25 | 78.81 ± 0.19 | 73.52 ± 0.14 | 86.30 ± 0.22 | 66.32 ± 0.27 | 72.81 ± 0.34 |
| UniGraph* [17] | 81.53 ± 0.18 | 86.07 ± 0.20 | 72.94 ± 0.33 | 78.47 ± 0.22 | 73.80 ± 0.48 | 86.44 ± 0.29 | 67.24 ± 0.31 | 73.51 ± 0.24 |
| GQT* [41] | 81.92 ± 0.26 | 85.59 ± 0.37 | 74.07 ± 0.47 | 77.52 ± 0.28 | 73.40 ± 0.11 | 85.66 ± 0.29 | 67.93 ± 0.24 | 73.22 ± 0.30 |
| GraphCLIP* [67] | 82.33 ± 0.27 | 84.95 ± 0.18 | 73.55 ± 0.20 | 78.14 ± 0.31 | 72.95 ± 0.17 | 84.92 ± 0.35 | 67.31 ± 0.51 | 73.40 ± 0.29 |
| FedGFM+ (Ours) | **83.79 ± 0.27** | **88.52 ± 0.31** | **76.31 ± 0.18** | **80.70 ± 0.28** | **75.25 ± 0.24** | **89.25 ± 0.13** | **69.39 ± 0.44** | **77.68 ± 0.22** |

## 5.1 Experimental Setup

To evaluate the effectiveness of FedGFM+, we conduct experiments on 8 benchmark graph datasets spanning a range of domains and covering three key tasks: node classification (Citation Networks: Cora, PubMed [56], and OGB-Arxiv [19]; Hyper-Link Networks: WikiCS [33]), edge classification (Knowledge Graphs: FB15K237 [37] and WN18RR [12]), and graph classification (Molecule Graphs: HIV, PCBA [47]). Each dataset is partitioned into 3 clients to simulate decentralized scenarios, and we report the average test performance (accuracy or AUC) across clients. We compare FedGFM+ against three baseline categories: (1) Isolated Supervised Models, trained independently on each client, including a linear layer, GCN, GAT, GraphSAGE, and GIN; (2) FL/FGL Approaches, including general-purpose methods like FedAvg and MOON, and task-specific methods such as FedSage+, Fed-PUB, FedGTA, FedTAD, FGSSL, FGGP, GCFL+, and FedStar; and (3) Federated Variants of centralized GFM training strategies (OFA, GFT, UniGraph, GQT, GraphCLIP). More experimental details are provided in Appendix C.

## 5.2 Performance Comparison (Answers for Q1)

To answer **Q1**, we compare FedGFM+ with a range of competitive baselines, evaluating each configuration over 3 independent runs without fixed seeds. As summarized in Table 2, FedGFM+ consistently achieves superior performance across all datasets and downstream tasks.

**Comparison with Isolated Supervised Learning.** FedGFM+ consistently outperforms supervised backbones, confirming its strong cross-domain and cross-task generalization. Specifically, it improves over the best baselines by at least 2.70% in node classification, 2.18% in edge classification, and 3.09% in graph classification, demonstrating superior transferability and robustness.

**Comparison with FL/FGL Methods.** As discussed in Section 1, existing FL/FGL methods are limited by data/task heterogeneity and reliance on task-specific information, restricting its training and evaluation scenarios. In contrast, as observed, FedGFM+ consistently outperforms by enabling broad cross- domain and task collaboration that captures general structural and semantic knowledge.

**Comparison with Federated Variants of Centralized GFM.** As observed, naive federated GFM models often suffer from knowledge entanglement, leading to them even below isolated supervised baselines (i.e., negative transfer). In contrast, FedGFM+ effectively addresses these issues via its design (i.e., AncDAI and AdaDPP), enabling efficient downstream adaptation.

## 5.3 Ablation Study (Answer for Q2)

To address **Q2**, we analyze FedGFM+s two key modules. **AncDAI** guides the initialization of learnable tokens in the global gVQ-VAE codebook, while **AdaDPP** is applied during fine-tuning to improve adaptability to domain- and task-specific variations. An ablation study on 8 datasets (Table 3) shows that removing both modules degrades performance. Notably, excluding AncDAI causes a larger drop than excluding AdaDPP, highlighting AncDAIs crucial role in reducing knowledge entanglement and boosting generalization. **In summary**, both are vital for FedGFM+s effectiveness.

Table 3: Ablation study results for FedGFM+. Node, edge, and graph classification datasets are marked in red, yellow, and blue, respectively.

| Dataset / Method | Cora | PubMed | OGB-arxiv | WikiCS | FB15K237 | WN18RR | HIV | PCBA |
|---|---|---|---|---|---|---|---|---|
| FedGFM+ w/o. AncDAI | 81.55 ± 0.22 | 85.56 ± 0.28 | 75.19 ± 0.19 | 78.05 ± 0.15 | 73.08 ± 0.31 | 87.61 ± 0.21 | 67.52 ± 0.11 | 74.81 ± 0.26 |
| FedGFM+ w/o. AdaDPP | 83.17 ± 0.18 | 87.42 ± 0.26 | 75.83 ± 0.27 | 77.64 ± 0.14 | 74.59 ± 0.26 | 88.19 ± 0.20 | 67.84 ± 0.29 | 76.72 ± 0.10 |
| FedGFM+ | **83.79** ± **0.27** | **88.52** ± **0.31** | **76.31** ± **0.18** | **80.70** ± **0.28** | **75.25** ± **0.24** | **89.25** ± **0.13** | **69.39** ± **0.44** | **77.68** ± **0.22** |

## 5.4 Sensitivity Analysis (Answer for Q3)

To address **Q3**, we perform a sensitivity analysis on key hyperparameters in FedGFM+. As a pre-trainingfine-tuning framework, it involves many hyperparameters; here we focus on those in our core modules. For AncDAI, we vary the number of learnable tokens in the global gVQ-VAE codebook. For AdaDPP, we vary the number of learnable prompts per client. Results are shown in Fig. 4: (a) AncDAI maintains stable performance under different codebook sizes, indicating robust domain initialization; (b) AdaDPP performs

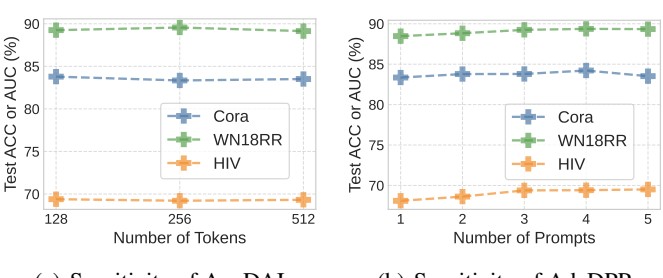

(a) Sensitivity of AncDAI  (b) Sensitivity of AdaDPP

Figure 4: Sensitivity analysis results for FedGFM+.

well with few prompts, and is insensitive to prompt number. Overall, FedGFM+ shows strong robustness to key hyperparameters.

## 6  Limitation

While FedGFM+ adopts randomly initialized encoders and decentralized optimization to mitigate privacy leakage, we acknowledge that the exchange of high-level representations (e.g., prototypes and prompts) may still expose partial semantic information. A thorough privacy analysis, including the investigation of potential leakage pathways and the development of a threat model, remains an important direction for future work. Incorporating formal privacy guarantees would further strengthen the robustness of our approach in practical federated settings.

## 7  Conclusion

This paper initiates the study of Federated Graph Foundation Models (FedGFM), aiming to train a unified graph model with domain and task generalization under decentralized settings. By integrating the complementary strengths of Federated Graph Learning (FGL) and centralized Graph Foundation Models (GFM) training strategies, FedGFM alleviates the limitations of both paradigms. Empirical analysis reveals a key challenge, knowledge entanglement, which limits the effectiveness of naive federated adaptations of centralized GFM training. To address this, we propose FedGFM+, a dual-perspective framework incorporating AncDAI and AdaDPP. Experimental results demonstrate the superior performance and generalization ability of FedGFM+.

## Acknowledgments

This work was supported in part by the National Key R&D Program of China under Grant 2023YFB3001900, in part by the Shenzhen Science and Technology Program under Grant KJZD20230923113901004, in part by the National Natural Science Foundation of China under Grant 62572501.

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

# Appendix: Table of Content

# A   More Related Works

**Graph Neural Networks.**   Earlier research on deep graph learning extends convolution to handle graphs [7] but comes with notable parameter counts. To this end, GCN [22] simplifies graph convolution by utilizing a 1-order Chebyshev filter to capture local neighborhood information. Moreover, GAT [39] adopts graph attention, allowing weighted aggregation. GraphSAGE [15] introduces a variety of learnable aggregation functions for performing message aggregation. Moreover, GIN [54] aims to preserve structural information maximally and theoretically proves its discriminative power matches the Weisfeiler-Lehman graph isomorphism test. Further details on GNN research can be found in surveys  [48, 64].

**Federated Graph Learning.**   Motivated by the success of federated learning in computer vision and natural language processing [55] and the demand for distributed graph learning, FGL has gained increasing attention. From the data and task perspectives, FGL studies are categorized into three settings: (1) **Graph-level FGL**, where each client collects multiple graphs for graph-level tasks, like graph classification. The main challenge is avoiding interference between clients' graph datasets, especially in multi-domain settings. For example, GCFL+ [53] introduces a GNN gradient pattern-aware technique for dynamic client clustering to reduce conflicts from structural and feature heterogeneity.  (2) **Subgraph-level FGL**, where each client holds a subgraph of a global graph for node-level tasks like node classification. The key challenges are *subgraph heterogeneity* and *missing edges*. Fed-PUB [3] addresses heterogeneity by enhancing local GNNs with random graph embeddings and personalized sparse masks for selective aggregation. FedGTA [26] encodes topology into smoothing confidence and graph moments to improve model aggregation.  Other studies [25, 20, 40, 66] also achieve strong results on this challenge. To address missing edges, Fed-Sage+ [62] integrates node representations, topology, and labels across subgraphs, training a neighbor generator to restore missing links and achieve robust subgraph-FL. Other works [8, 57, 61] also excel in this area. (3) **Node-level FGL**, where each client collects one or multiple ego-networks for node- and edge-level tasks. From the perspective of data format and task, Node-level FGL can be seen as a special case of Subgraph-level FGL. Notably, the application scenarios of Node-level FGL usually involve strict privacy constraints, and representative methods include FedEgo [63]. Detailed insights into FGL research are available in surveys [14, 60, 13] and benchmark studies [16, 45, 27].

**Language-Oriented GFMs** [28, 65, 17, 23].  These approaches transform graph structures into linearized textual sequences by encoding nodes and edges using syntactically structured templates. The resulting representations can then be processed by token-based encoderstypically LLMsthat are pre-trained on vast corpora of natural language. This approach allows for seamless integration with existing LLM infrastructure and leverages the powerful contextual understanding capabilities developed through natural language processing (NLP). In more detail, during the pre-training phase, these models optimize the parameters of the embedding functionoften realized as an LLMthrough conventional NLP objectives such as next-token prediction or masked language modeling. These objectives encourage the model to learn coherent semantic representations from the flattened graph text, effectively transferring linguistic inductive biases to graph representation learning. However, despite their ability to inherit the expressive power of LLMs, language-oriented GFMs face intrinsic limitations. The transformation from graph to text inevitably introduces information loss, especially concerning structural properties such as node connectivity and subgraph patterns. Moreover, this flattening process may distort the original graph topology in ways that are not easily reversible, thereby affecting downstream tasks that rely on accurate structural reasoning. Additionally, scalability becomes a concern due to the growing length of textual sequences with increasing graph size, which may lead to inefficiencies in both computation and memory usage.

**Graph-Oriented GFMs** [50, 59, 31, 9, 52, 29, 44, 42, 43, 46, 67, 35, 58]. These approaches aim to preserve both the semantic richness of textual attributes and the integrity of graph topology through purpose-built architectures. These models typically adopt a hybrid design, wherein a frozen LLM is used to extract high-quality textual embeddings from node and edge features, while a trainable GNN component handles the aggregation and propagation of information across the graph structure. This dual-component architecture enables the model to benefit from the strong language understanding capabilities of LLMs without compromising the fidelity of graph structure. The GNN backbone ensures that topological relationships are explicitly modeled, allowing for effective message passing and relational reasoning. During the pre-training stage, graph-oriented GFMs often incorporate self-supervised learning strategies, such as graph reconstruction or contrastive learning objectives, which

help the model capture invariant and transferable representations across diverse domains. These tasks encourage the model to learn a unified representation space where both textual and structural semantics are coherently aligned, leading to better generalization on downstream tasks involving heterogeneous graph data. By preserving the native structure of graphs and leveraging the representational power of modern neural architectures, graph-oriented GFMs offer a promising direction toward building robust and scalable foundation models for graph-centric machine learning.

## B  Theoretical Proof

In this section, we provide theoretical analysis for the distinguishability of domain prototypes under random initialization (Theorem. B.1) and the semantic separability of gVQ-VAE codebooks initialized via AncDAI (Theorem. B.2).

**Theorem B.1** (Domain Prototype Distinguishability). *Let $G^a = (\mathcal{V}^a, \mathcal{E}^a)$ and $G^b = (\mathcal{V}^b, \mathcal{E}^b)$ denote local graphs from two clients belonging to different domains, with node features $\mathbf{X}^a, \mathbf{X}^b \in \mathbb{R}^{n \times d}$ and adjacency matrices $\mathbf{A}^a, \mathbf{A}^b \in \mathbb{R}^{n \times n}$. Let $f_\theta^{glb}$ be the parameters of a randomly initialized $L$-layer global GNN-Encoder, which is broadcast to all clients for local initialization. The domain prototype is computed with Eqs. (3) and (4):*

$$\mathbf{p}^a = \frac{1}{n} \sum_{i=1}^n f_\theta^{glb}(\mathbf{A}^a, \mathbf{X}^a)_i, \quad \mathbf{p}^b = \frac{1}{n} \sum_{i=1}^n f_\theta^{glb}(\mathbf{A}^b, \mathbf{X}^b)_i. \tag{8}$$

*Then, there exists a constant $\alpha > 0$, whose value depends on the architecture and depth $L$ of GNN-Encoder), such that:*

$$\mathbb{E}_\theta \left[ \left\| \mathbf{p}^a - \mathbf{p}^b \right\|_2^2 \right] \geq \alpha \cdot \left( \left\| \mathbf{X}^a - \mathbf{X}^b \right\|_F^2 + \left\| \mathbf{A}^a - \mathbf{A}^b \right\|_F^2 \right). \tag{9}$$

*Proof.* Let $\mathbf{z}_i^a = f_{\boldsymbol{\theta}}^{\mathrm{glb}}(\mathbf{A}^a, \mathbf{X}^a)_i$ and $\mathbf{z}_i^b = f_{\boldsymbol{\theta}}^{\mathrm{glb}}(\mathbf{A}^b, \mathbf{X}^b)_i$ denote the representations of node $i$ obtained from a frozen GNN applied to graphs $a$ and $b$, respectively. Here, the GNN parameters $\boldsymbol{\theta}$ are randomly initialized and held fixed. Under this setting, the GNN's computations can be interpreted as performing random but deterministic linear transformations and message passing operations. Leveraging the linearity of expectation and the independence of random initialization, the expected squared Euclidean distance between the resulting node prototypes can be expressed as:

$$\mathbb{E}_\theta \left[ \left\| \mathbf{p}^a - \mathbf{p}^b \right\|_2^2 \right] = \mathbb{E}_\theta \left[ \left\| \frac{1}{n} \sum_{i=1}^n \left( \mathbf{z}_i^a - \mathbf{z}_i^b \right) \right\|_2^2 \right] \tag{10}$$

$$\geq \frac{1}{n^2} \sum_{i=1}^n \mathbb{E}_\theta \left[ \left\| \mathbf{z}_i^a - \mathbf{z}_i^b \right\|_2^2 \right] \tag{11}$$

$$\geq \alpha \cdot \left( \left\| \mathbf{X}^a - \mathbf{X}^b \right\|_F^2 + \left\| \mathbf{A}^a - \mathbf{A}^b \right\|_F^2 \right). \tag{12}$$

$\square$

**Theorem B.2** (Semantic Separability of AncDAI-Initialized Codebook). *Let $\{\mathbf{p}^k\}_{k=1}^K$ be the set of domain prototypes uploaded from $K$ clients. For each prototype $\mathbf{p}^k$, we generate a set of perturbed vectors via Eq. 5:*

$$\tilde{\mathbf{p}}_i^k = \mathbf{p}^k + \sigma \boldsymbol{\epsilon}_i, \quad \boldsymbol{\epsilon}_i \sim \mathcal{N}(\mathbf{0}, \mathbf{I}), \quad i = 1, \ldots . H. \tag{13}$$

*Let $\mathcal{C}^{perturb}$ and $\mathcal{C}^{rand}$ be codebooks constructed respectively from perturbed prototypes and from standard Gaussian initialization. Then for any two domains $a \neq b$ and respective node embeddings $\mathbf{z}^a$, $\mathbf{z}^b$ (drawn from $f_\theta(\mathbf{A}, \mathbf{X})$), we have:*

$$\mathbb{P}\left[ code(\mathbf{z}^a; \mathcal{C}^{perturb}) \neq code(\mathbf{z}^b; \mathcal{C}^{perturb}) \right] \geq \mathbb{P}\left[ code(\mathbf{z}^a; \mathcal{C}^{rand}) \neq code(\mathbf{z}^b; \mathcal{C}^{rand}) \right], \tag{14}$$

*i.e., the perturbation-initialized codebook yields higher domain-level separability.*

*Proof.* We adopt a quantization function based on cosine similarity:

$$code(\mathbf{z}; \mathcal{C}) = \arg \max_{\mathbf{c} \in \mathcal{C}} \frac{\mathbf{z}^\top \mathbf{c}}{\|\mathbf{z}\|_2 \|\mathbf{c}\|_2},$$

which assigns each embedding to the codebook vector with the smallest angular distance.

Assume that the domain prototypes $\{\mathbf{p}^k\}_{k=1}^K$ satisfy a minimal angular separation:

$$\min_{a \neq b} \arccos \left( \frac{(\mathbf{p}^a)^\top \mathbf{p}^b}{\|\mathbf{p}^a\|_2 \|\mathbf{p}^b\|_2} \right) \geq \delta > 0.$$

The perturbed codebook $\mathcal{C}^{\text{perturb}}$ is formed by adding isotropic Gaussian noise $\sigma \epsilon$ to each prototype, with $\epsilon \sim \mathcal{N}(\mathbf{0}, \mathbf{1})$. For sufficiently small $\sigma$, the perturbations preserve the cluster structure, yielding distinct codebook clusters separated by angles close to $\delta$.

Node embeddings $\mathbf{z}^a$ and $\mathbf{z}^b$ sampled from different domains concentrate in neighborhoods around their respective prototypes. Formally, with high probability,

$$\arccos \left( \frac{(\mathbf{z}^a)^\top \mathbf{p}^a}{\|\mathbf{z}^a\|_2 \|\mathbf{p}^a\|_2} \right) \leq \epsilon, \quad \arccos \left( \frac{(\mathbf{z}^b)^\top \mathbf{p}^b}{\|\mathbf{z}^b\|_2 \|\mathbf{p}^b\|_2} \right) \leq \epsilon,$$

for some small $\epsilon > 0$. Then. by the triangle inequality on the unit sphere,

$$\arccos \left( \frac{(\mathbf{z}^a)^\top \mathbf{z}^b}{\|\mathbf{z}^a\|_2 \|\mathbf{z}^b\|_2} \right) \geq \delta - 2\epsilon,$$

which implies that embeddings from distinct domains remain well-separated.

Therefore, the probability that $\mathbf{z}^a$ and $\mathbf{z}^b$ are assigned to the same codeword under $\mathcal{C}^{\text{perturb}}$ is bounded above by the probability that perturbations cause cluster overlap, which is small for sufficiently small $\sigma$. In contrast, a random codebook $\mathcal{C}^{\text{rand}}$ sampled isotropically from a standard Gaussian lacks such separation, and embeddings from different domains have a higher probability of being assigned the same codeword. Thus, we combines these observations and proof that:

$$\mathbb{P}\big[\text{code}(\mathbf{z}^a; \mathcal{C}^{\text{perturb}}) \neq \text{code}(\mathbf{z}^b; \mathcal{C}^{\text{perturb}})\big] \geq \mathbb{P}\big[\text{code}(\mathbf{z}^a; \mathcal{C}^{\text{rand}}) \neq \text{code}(\mathbf{z}^b; \mathcal{C}^{\text{rand}})\big].$$

$\square$

# C  More Detailed Experimental Setup

## C.1  Dataset

Table 4: The statistics of evaluated datasets in our experiments.

| Dataset | Domain | Task | # Graphs | Avg. #Nodes | Avg. #Edges | # Classes |
|---------|--------|------|----------|-------------|-------------|-----------|
| Cora | Citation | Node | 1 | 2,708 | 10,556 | 7 |
| PubMed | Citation | Node | 1 | 19,717 | 44,338 | 3 |
| Arxiv | Citation | Node | 1 | 169,343 | 1,166,243 | 40 |
| WikiCS | Hyper link | Node | 1 | 11,701 | 216,123 | 10 |
| FB15K237 | Knowledge | Link | 1 | 14,541 | 310,116 | 237 |
| WN18RR | Knowledge | Link | 1 | 40,943 | 93,003 | 11 |
| PCBA | Molecule | Graph | 437,929 | 26.0 | 28.1 | 128 |
| HIV | Molecule | Graph | 41,127 | 25.5 | 27.5 | 2 |

We utilize 8 datasets from various domains and tasks, as detailed in Table 4.

## C.2  Data Processing

Our data processing process can be illustrated as Fig. 5, consisting of two steps: **Step 1: Language Encoding.** We use Sentence-Bert [34] to uniformly encode text attribute graph datasets in different fields to uniformly convert node and edge text into 768-dimensional vectorized representations; and **Step 2: Data Decentralization Simulation.** Real-world graph data is inherently collected by multiple institutions, resulting in naturally decentralized data distributions. Prior studies in FGL categorize such decentralization into three canonical levels [16]: (1) node-level, where each client maintains ego-networks extracted from a global graph; (2) subgraph-level, where each client collects

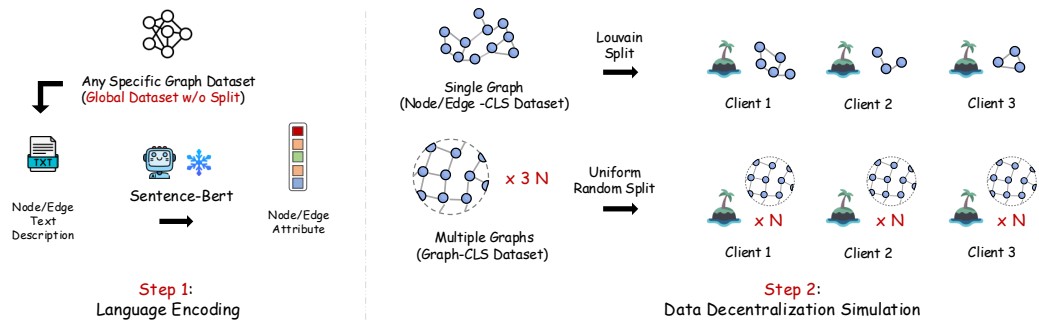

Figure 5: Data processing pipeline to simulate decentralized multi-domain and task graphs.

a local subgraph induced from a broader graph topology; and (3) graph-level, where each client independently gathers a set of graphs from a larger collection. Notably, the node-level setting can be regarded as a special case of subgraph-level decentralization. Hence, we focus on the latter two in this work. Specifically, under the subgraph-level setting, the implicit global graph $G = (\mathcal{V}, \mathcal{E})$ has multiple substructures independently collected by different clients. The $k$-th client locally collects a subgraph $G_k = (\mathcal{V}_k, \mathcal{E}_k)$ such that $\mathcal{V}_k \subsetneq \mathcal{V}$ and $\mathcal{E}_k \subsetneq \mathcal{E}$; Under the graph-level setting, the $k$-th client independently collects a subset of graphs $\mathcal{S}_k$ from an implicit broader collection $\mathcal{S} = \{G_i\}_{i=1}^{M}$, i.e., $\mathcal{S}_k \subsetneq \mathcal{S}$. To simulate these decentralized scenarios in our experiments, we adopt two partitioning strategies: the Louvain algorithm [5] for simulating subgraph-level decentralization, and random allocation for graph-level decentralization, both of which is widely used in various FGL studies [27].

Finally, the default train/validation/test splits used in the fine-tuning stage are summarized in Table. 5. Notably, due to the distributed nature of federated settings, the training set proportion is typically much higher than in centralized graph learning paradigms. This splitting strategy has been widely adopted in prior works [27].

Table 5: Train/Validation/Test splits for different datasets

| Dataset | Train Split | Validation Split | Test Split |
|---|---|---|---|
| Cora | 5% | 20% | 40% |
| PubMed | 60% | 20% | 20% |
| WikiCS | 80% | 10% | 10% |
| Arxiv | 80% | 10% | 10% |
| WN18RR | 80% | 10% | 10% |
| FB15k237 | 80% | 10% | 10% |
| ChemHIV | 80% | 10% | 10% |
| ChemPCBA | 80% | 10% | 10% |

### C.3 Baselines

Since this paper is the first to explore FedGFM, we transfer baselines from adjacent fields. Specifically, in our experiments, we evaluate 20 baselines, which can be summarized into 3 categories. The detailed descriptions of these baselines are as follows:

**(1) Isolated Supervised Learning.** These methods train individual supervised models on each client without federated communication. They serve as a reference for evaluating negative transfer and the benefits of federated learning. The models in this category include a linear layer, GCN [22], GAT [39], GraphSAGE [15], and GIN [54];

**GCN** [22] is a classical model in graph neural networks, which captures graph structure through spectral convolutions based on the normalized graph Laplacian. By aggregating information from neighboring nodes, it enables efficient node classification and handles graph data in a computationally effective manner. The use of the Laplacian matrix simplifies the convolution operation, making it a foundational approach in graph representation learning.

**GAT** [39] draws inspiration from the success of attention mechanisms in natural language processing, introducing a novel graph attention mechanism that allows nodes to dynamically focus on the most relevant neighbors. This attention-based aggregation enables more adaptive learning.

**GraphSAGE** [15] extends graph neural networks by introducing a sampling-based message passing mechanism, which allows for scalable neighborhood aggregation. This approach is particularly well-suited for inductive learning, as it can efficiently generalize to unseen nodes by sampling a fixed-size neighborhood during training. The use of different learnable aggregation functions further enhances scalability, enabling the model to handle large graphs effectively.

**GIN** [54] is designed to preserve graph structural information and has been shown to be as expressive as the Weisfeiler-Lehman graph isomorphism test in distinguishing graph structures. Notably, GIN is usually more suitable for graph-level tasks.

**(2) FGL Approaches.** We evaluate various representatives FL/FGL baselines, including two FL methods desinged for FL with vision tasks (FedAvg [32], MOON [24]), and subgraph-level FGL techniques (FedSage+ [62], Fed-PUB [3], FedGTA [26], FedTAD [66], FGSLL [20], FGGP [40]) and graph-level FGL methods (GCFL [53] and FedStar [36]). The detailed descriptions of these baselines are as follows:

**FedAvg** [32] is a simple yet effective method in FL for the vision and language field, enabling decentralized model training while preserving data privacy. A central server distributes the global model to clients for local updates. The server then aggregates the clients' local models to form a new global model, which is broadcast to all clients to update their local models in the next round.

**MOON** [24] is a representative FL method originally developed for the vision domain. It leverages contrastive learning at the model level to align local and global representations, thereby mitigating performance degradation caused by data heterogeneity across clients.

**FedSage+** [62] integrates node features, link structures, and labels using a GraphSAGE [15] model with FedAvg [32] for FGL over local subgraphs (i.e., subgraph-level FGL). It also introduces a neighbor generator to handle cross-client missing links, improving robustness and ensuring a more comprehensive graph representation.

**Fed-PUB** [3] is a personalized subgraph-level FGL framework that improves local GNNs without relying on a global model. It measures inter-client similarity using functional embeddings derived from random graph inputs, enabling weighted aggregation at the server. A client-specific sparse mask further guides personalized parameter updates, facilitating subgraph-aware local adaptation.

**FedGTA** [26] integrates large-scale graph learning into FGL by having clients encode topology and node attributes, compute local smoothing confidence and mixed feature moments, and share them with the server. The server aggregates personalized models using smoothing confidence as aggregation weights.

**FedTAD** [66] is a subgraph-level FGL method that computes topology-aware node embeddings to estimate class-wise knowledge reliability. This guidance enables the server to perform data-free knowledge distillation, transferring reliable knowledge from local clients to the global model.

**FGSSL** [20] is a subgraph-level FGL technique, which addresses client drift by aligning node-level semantics and preserving graph-level structures. It employs contrastive objectives to align nodes of the same class while separating different classes, and distills global relational knowledge into local models via similarity distributions.

**FGGP** [40] is a subgraph-level FGL approach, which decomposes the global model into two tiers connected via prototypes. At the classifier level, class prototypes replace traditional classifiers for better discriminability; at the feature level, contrastive learning injects global knowledge into prototypes to enhance generalization.

**GCFL+** [53] is a graph-level FGL framework that clusters clients based on GNN gradient patterns to address structural and feature heterogeneity. It further improves stability through gradient sequence-based clustering using dynamic time warping, enhancing both clustering quality and robustness.

**FedStar** [36] enables graph-level FGL by decoupling structure and feature learning. Clients share domain-invariant structural embeddings via an independent encoder, while learning personalized features locally, reducing feature misalignment and improving transferability.

**(3) Federated Variants of Centralized GFM Approaches.** These baselines adapt state-of-the-art centralized GFM training strategies to the federated setting. Specifically, we include OFA [29], GFT [43], UniGraph [17], GQT [41], and GraphCLIP [67]. In their original centralized versions, these methods perform pre-training on all available data at a central learning system using self-supervised objectives. Their federated counterparts distribute this pre-training process across clients. **Specifically, for our experiments,** in each communication round of the pre-training phase, each local client deploys the corresponding framework based on its own local data and performs 2 epoch optimization. Subsequently, all trainable parameters will be uploaded to the server, and the parameters will be averaged to obtain the global model, which will be broadcast to all clients as the starting point for the next round of local optimization.

**OFA**$^*$ [29] is a representative training paradigm for GFM, aiming to learn generalizable representations over cross-domain and cross-task textual attributed graphs. It first standardizes the description of nodes and edges via carefully designed language model prompts, transforming any textual attributed graph into a unified vectorized representation. Additionally, OFA introduces NODES-OF-INTEREST prompts to unify various graph tasks within a single modeling framework.

**GFT**$^*$ [43] treats computation trees derived from message passing as transferable patterns over graphs. Based on this insight, it adopts a gVQ-VAE architecture to map computation trees into discrete codebook representations. Through self-supervised reconstruction on cross-domain graphs during pre-training, it learns a generalizable GFM with strong cross-graph transferability.

**UniGraph**$^*$ [17] is a GFM training framework that encodes heterogeneous graphs, including those without inherent textual features, into unified textual representations to support cross-domain transferability. It adopts a cascaded architecture of language models and GNNs to jointly capture semantic and structural information. UniGraph further introduces a Masked Graph Modeling objective for large-scale self-supervised pre-training and applies graph instruction tuning with LLMs to enhance zero-shot and few-shot generalization.

**GQT**$^*$ [41] introduces a novel graph quantized tokenizer that decouples tokenizer training from Transformer training, leveraging multi-task graph self-supervised learning to produce robust and generalizable graph tokens. By using the residual Vector Quantization technique, GQT learns hierarchical discrete tokens, reducing memory requirements and enhancing generalization.

**GraphCLIP**$^*$ [67] addresses key challenges in text-attributed graphs, including heavy reliance on label information and limited cross-domain transferability. It introduces a self-supervised contrastive pretraining method using graph-summary pairs curated with the help of LLMs. By leveraging invariant learning, GraphCLIP enhances zero-shot transferability and proposes a graph prompt tuning technique for few-shot learning, mitigating catastrophic forgetting.

## C.4  Model Architecture

For **Isolated Supervised Learning Methods**, we adopt a two-layer architecture with 64 hidden units. For **FL/FGL Methods**, if a method does not specify a custom architecture, we select the backbone based on the downstream task: GraphSAGE is used for node and edge classification, while GIN is employed for graph classification. For **Federated Variants of Centralized GFM Methods**, we follow the backbone choices reported in the original papers. For **FedGFM+**, we employ a gVQ-VAE as the backbone for both client-side local models and the server-side global model. The encoder is a 2-layer GraphSAGE-based graph convolutional network that jointly encodes node and edge features from the input graph $G = (V, E)$. All layers including input, hidden, and output are set to 768 dimensions, matching the Sentence-BERT [34] representations of node and edge attributes. The encoder outputs node embeddings $Z \in \mathbb{R}^{N \times 768}$, where $N$ is the number of nodes. These embeddings are then quantized via a multi-head gVQ-VAE codebook using cosine similarity for nearest-neighbor retrieval. The codebook comprises 4 heads, each containing 128 learnable tokens. A shared linear projection is applied to aggregate the multi-head outputs into the final quantized representation. In addition to the backbone network, FedGFM+ also introduces multiple light-weight learnable graph hints for each client. By default, we learn 3 local graph prompts for each client. Finally, for task-specific heads used during GFM fine-tuning, we follow the original design if specified in the corresponding dataset paper. Otherwise, for node classification, we apply a single-layer MLP to predict node labels from node embeddings; for edge classification, we average the embeddings of the two nodes to form the edge representation and apply a single-layer

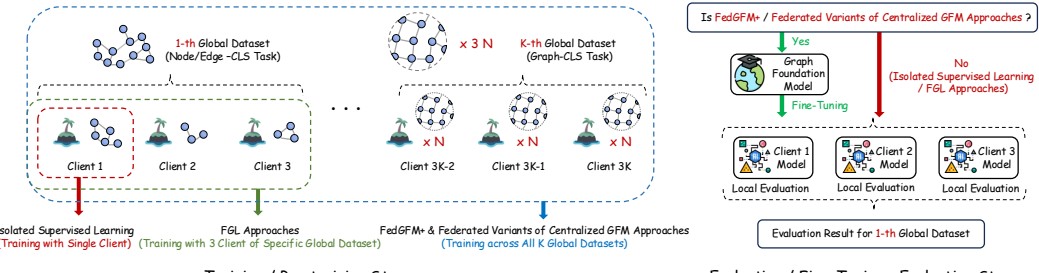

Figure 6: Illustration of the pipeline about the traing and evaluation stage for isolated supervised learning, FGL approaches, FedGFM+, and federated variants of centralized GFM approaches.

MLP; for multi-task graph classification, we perform mean pooling over node embeddings to obtain a graph-level representation, which is fed into a MLP to predict binary labels for each tasks.

## C.5 Training and Evaluation Illustration

We illustrate the training and evaluation processes for all baselines and FedGFM+ in Fig. 6, with detailed descriptions as follows:

**Training / Pretraining Stage.** For the **Isolated Supervised Learning Baselines**, each client trains a model independently from scratch, using only its own local graph(s), without any collaboration or information exchange. For the **FL/FGL Baselines**, we run a FL/FGL algorithm among every 3 clients from the same global dataset. For example, the Cora dataset is split using the Louvain algorithm into clients 1, 2, and 3, and subgraph-level FGL algorithms such as FedGTA are then applied among these clients. Notably, as mentioned in Sec. 1, due to the heterogeneity of data and tasks, most FL/FGL algorithms can only be simulated among different shards of the same dataset. Moreover, existing FGL algorithms cannot be applied simultaneously to the three tasks of node classification, edge classification, and graph classification. For **FedGFM+ and Federated Variants of Centralized GFM Baselines**, all clients participate in federated pre-training together, which enables extensive collaboration among graph datasets from multiple fields.

**Evaluation / Fine-Tuning + Evaluation Stage.** For each dataset, we evaluate the performance on the test sets of the three clients associated with it, and report the mean and variance of the resulting metrics. For node and edge classification tasks, we use Accuracy (ACC) as the evaluation metric, while for graph classification tasks, we adopt the Area Under the Receiver Operating Characteristic Curve (AUC-ROC). To assess the performance of each individual client under different settings, we follow distinct evaluation protocols. For Isolated Supervised Learning baselines, we directly evaluate each clients local model without any collaboration. For FL and FGL baselines, we evaluate each clients model after training the global model for two communication rounds. For FedGFM+ and the federated variants of centralized GFM baselines, we first attach a task-specific header and then fine-tune the model using each clients local graph before evaluation.

## C.6 Hyperparameters

For **Isolated Supervised Learning Baselines**, we perform 1,000 epochs of local training with early stopping based on validation performance. For **FL/FGL Baselines**, we conduct 100 communication rounds, where each round includes 2 local training epochs. We use the Adam optimizer with a learning rate of $1 \times 10^{-2}$, weight decay of $5 \times 10^{-4}$, and dropout rate of $0.5$. For **federated variants of centralized GFM Baselines**, we adopt the hyperparameter configurations reported in their original papers whenever available. When unspecified, we employ automated hyperparameter optimization using the Optuna framework [2]. Federated pre-training is carried out for 50 communication rounds, each consisting of 2 local pre-training epochs. For our proposed **FedGFM+** framework, we fix the learning rate for pre-training to $1 \times 10^{-4}$. During fine-tuning, we perform a grid search over learning rates in $\{10^{-5}, 10^{-4}, 10^{-3}, 10^{-2}, 10^{-1}\}$ for each dataset. The weight decay is fixed to $5 \times 10^{-4}$, and the batch size is set to 1,024. Federated pre-training is conducted for 25 communication rounds, with 2 local epochs per round.

## C.7 Experimental Environment

The experimental machine is an Intel(R) Xeon(R) Gold 6240 CPU @ 2.60GHz and NVIDIA A100 with 80GB memory and CUDA 12.4. The operating system is Ubuntu 22.04.5 with 251GB memory.

# D Few-shot Learning Results

We perform a few-shot evaluation across a range of downstream tasks. Specifically, for node and edge classification tasks, we constrain the number of labeled samples per class to at most 2. For graph classification tasks, however, we do not report few-shot performance, as each graph instance is associated with multi-dimensional labels, making few-shot evaluation non-trivial. The experimental results are summarized as Table. 6

Table 6: 2-shot Performance comparison of FedGFM+ and baselines. '*' denotes federated variants of centralized GFM. 'N/A' denotes task inapplicability. Node and edge classification datasets are marked in red and yellow, respectively.

| Dataset / Method | Cora | PubMed | OGB-arxiv | WikiCS | FB15K237 | WN18RR |
|---|---|---|---|---|---|---|
| OFA$^*$ [29] | 54.31 $\pm$ 0.18 | 45.29 $\pm$ 0.26 | 20.56 $\pm$ 0.42 | 40.05 $\pm$ 0.10 | 19.72 $\pm$ 0.33 | 31.28 $\pm$ 0.20 |
| GFT$^*$ [43] | 52.16 $\pm$ 0.39 | 44.71 $\pm$ 0.10 | 18.31 $\pm$ 0.22 | 37.42 $\pm$ 0.56 | 17.49 $\pm$ 0.24 | 29.55 $\pm$ 0.41 |
| UniGraph$^*$ [17] | 54.22 $\pm$ 0.27 | 46.41 $\pm$ 0.50 | 19.88 $\pm$ 0.15 | 39.46 $\pm$ 0.17 | 18.45 $\pm$ 0.36 | 31.53 $\pm$ 0.20 |
| GQT$^*$ [41] | 52.45 $\pm$ 0.18 | 45.28 $\pm$ 0.26 | 20.10 $\pm$ 0.31 | 39.25 $\pm$ 0.42 | 20.40 $\pm$ 0.18 | 30.08 $\pm$ 0.14 |
| GraphCLIP$^*$ [67] | 55.31 $\pm$ 0.12 | 44.25 $\pm$ 0.36 | 20.39 $\pm$ 0.17 | 38.58 $\pm$ 0.16 | 20.58 $\pm$ 0.28 | 31.42 $\pm$ 0.45 |
| FedGFM+ (Ours) | **58.33** $\pm$ **0.42** | **50.19** $\pm$ **0.23** | **21.34** $\pm$ **0.15** | **43.35** $\pm$ **0.39** | **21.94** $\pm$ **0.17** | **33.64** $\pm$ **0.42** |

As observed, FedGFM+ consistently outperforms naive federated adaptations of centralized GFM training strategies across all evaluated settings. By integrating the AncDAI and AdaDPP modules, FedGFM+ effectively constructs domain-aware semantic priors that enhance generalization to downstream tasks in heterogeneous domains, even with limited fine-tuning labels. Despite these gains, it is important to note that FedGFM+ still falls short of its own performance under scenarios with abundant labeled data.

