# OpenReview forum: "Towards Effective Federated Graph Foundation Model via Mitigating Knowledge Entanglement"
_NeurIPS.cc/2025/Conference — NeurIPS 2025 poster_

### Official Review · Reviewer_PsLJ · 2025-07-01

**Clarity:** 2
**Significance:** 1
**Originality:** 2
**Rating:** 3
**Confidence:** 4

**Summary:**

This paper is trying to solve question containing federated graph learning and graph foundation models. To address the knowledge entanglement challenge, FedGFM+ is proposed to solve it from the global and the local perspective. The authors conduct cross-task and cross-domain experiments to demonstrate the effectiveness of the approach.

**Questions:**

1, Could you please clarify the motivation of the proposed approach? From the methodology to the applications.

2, Could you please identify which individual technique is innovatively proposed in this paper?

3, Although I read that you categorize the baselines roughly, could you please clarify a little bit more about how you select the baselines?

**Ethical Concerns:**

["NO or VERY MINOR ethics concerns only"]

**Final Justification:**

The authors have addressed my concerns of baseline selection. But I am concerned about the KEY novelty of each module and the potential issues of privacy leakage. I do not think I can raise my score to positive at this moment.

**Limitations:**

1, I suggest the authors to think deeper about the KEY motivation of the proposed approach and present it more clearly.

2, Also, one or two focused points deserves to be explored comprehensively.

**Quality:**

2

**Strengths And Weaknesses:**

Strengths:

(1) Figure 1 is a good diagram to demonstrate the comparison.

(2) They conduct extensive experiments to verify their framework.

Weakness:

(1) The motivation is not clear. The combination of two terms seems not convincing as the motivation.

(2) The authors try hard to include too many aspects in one paper, which confuses the reader a lot. I understand that the data heterogeneity, task heterogeneity, multi-domain data isolation, cross-silo storage and computation neglect are the limitations for the two terms respectively. But the statements from Line 64 to 78 are far-fetched and strained.

(3) This framework contains multiple modules. But most of them are not novel if we treat them individually, for example, the prototype and personalized prompt.

---

> ### Author Rebuttal · Authors · 2025-07-26
>
> We sincerely thank you for your comments, and hope to address your concerns as follows:
>
>
> ---
>
> **Question 1**: The motivation of the FedGFM paradigm (combining FGL and GFM) is not convincing.
>
> We respectfully highlight that FedGFM is not a superficial field combination, but a practical paradigm that leverages the complementary strengths of FGL and GFM.
>
> - The advantage of FGL is that it can achieve cross-silo data collaboration, computing and storage resource collaboration under the scenario of privacy protection. However, its disadvantage is that the scope of collaboration is extremely limited (across small number of fields or even different shards of a single dataset) due to the data heterogeneity (including features, space, topology) and task heterogeneity (bound to specific model architecture, algorithm process, and optimization goals).
>
> - On the contrary, the advantage of GFM is that the language model encoding unifies the data format, while the pre-training mechanism is task-format-agnostic. The disadvantage is that the actual training of GFM faces the difficulty of centrally collecting multi-domain datasets (graphs from different domains are collected by different institutions and are not allowed to be made centralized and public) and the opportunity cost of wasting multi-party cross-silo computing and storage resources.
>
> **Why centralized GFMs are not enough to fully meet real-world needs**: Existing centralized GFM studies are typically trained on datasets from a small range of domains (e.g., 4 domains for OFA[1] and GFT[2]), which is significantly fewer than those used in vision or NLP foundation models. However, building a generalized graph foundation model likely requires access to graphs from a much broader range of domains. In reality, such data (e.g., financial networks, disease graphs) are often privacy-sensitive and siloed across institutions, where legal, ethical, and commercial constraints prohibit direct data sharing. As a result, aggregating these graphs for centralized GFM training is not only impractical but often infeasible.
>
> FedGFM addresses this limitation by aligning GFM’s task-agnostic modeling capabilities with FGL’s decentralized training infrastructure, enabling scalable foundation model pre-training across siloed, heterogeneous, and sensitive domains, which is the core motivation behind FedGFM.
>
> ---
>
> **Question 2**: The various modules of the proposed framework FedGFM+ lack technical novelty.
>
> We respectfully argue that the core novelty of this work lies in **paradigm innovation**: to the best of our knowledge, this is the first work to identify and systematically characterize the complementary relationship between FGL and GFM, leading to the proposal of a new paradigm, FedGFM. We further conduct an in-depth empirical analysis to uncover a paradigm-specific challenge named knowledge entanglement.
>
> Within this new paradigm, the **novelty of technical design** should be evaluated in terms of how our method addresses the above challenge. While certain components like prototypes and personalized prompts have been widely used in various machine learning research, our design motivations and functionalities are uniquely aligned with the needs of FedGFM, rather than being direct adaptations from any FGL, GFM, or VQ-VAE studies. Below, we provide a detailed clarification of the novel aspects of each module:
>
>
> - ***Anchor-Based Domain-Aware Initialization (AncDAI)***: In FedGFM, where centralized optimization of multi-domain knowledge is infeasible, clients from different domains may encode distinct semantics into the same codebook tokens. When these locally updated codebooks are aggregated on the server, such semantic conflicts can compromise the global representation quality. To mitigate this, AncDAI leverages each client’s local domain prototype as a semantic anchor to initialize the global codebook, encouraging early separation of domain-specific semantics and reducing conflict during aggregation. **This design is unique to FedGFM paradigm**, as existing GFM or VQ-VAE models operate in centralized settings, where token semantics can be aligned through joint optimization, eliminating the need for such initialization strategies.
>
>
>
>
>
> - ***Adaptive Domain-Sensitive Prompt Pool (AdaDPP)***: To mitigate knowledge entanglement in FedGFM from the local perspective, AdaDPP learns domain-specific prompts during pre-training and adaptively injects them during fine-tuning. These prompts provide domain-aware guidance that modulates the shared encoder’s behavior, enabling localized adaptation while reducing cross-domain interference. Unlike prior GFM or FGL methods that rely on a shared encoder without domain-specific adaptation, AdaDPP introduces explicit domain-aware modulation to preserve semantic boundaries across domains.
>
> ---
>
> **Question 3**: Need more explanation about the baseline selection strategy and the reason behind it.
>
> We sincerely apologize for any possible oversights. Here we provide further explanations about the reasons for the setting and selection of each category baseline:
>
> - ***Isolated Supervised Learning***: These models are trained independently on each client without collaboration, serving as the performance lower bound under non-federated and non-pre-training settings. They help assess the benefit of FedGFM over traditional per-graph training. Model choices are also aligned with task types: GCN, GAT, and GraphSAGE for node/edge classification, and GIN for graph classification.
>
>
>
> - ***FL/FGL Methods***: These approaches support only limited collaboration across domains and are often tailored to specific downstream tasks, limiting their generality. In contrast, FedGFM enables broad, scalable collaboration across heterogeneous domains and tasks. The performance comparison with FL/FGL baselines further highlights the effectiveness of generalized, task-agnostic pretraining over narrow, task-specific federated learning.
>
> - ***Federated Variants of Centralized GFM***: These methods adapt centralized GFM architectures to federated settings by performing multi-round parameter aggregation across clients. While they enable federated pretraining, they do not account for the knowledge entanglement challenge arising from cross-domain heterogeneity. In contrast, FedGFM+ is explicitly designed to mitigate such entanglement, and comparisons with these baselines highlight its superior ability to preserve domain-specific semantics and improve performance under the FedGFM paradigm.
>
>
> Finally, we kindly remind you that detailed descriptions of each baseline catagories and specific methods can be found in Appendix C.3. We promise you that we will add these supplementary content to the appendix during the revision phase.
>
>
> ---
>
> **References**
>
> [1] Liu, H., Feng, J., Kong, L., Liang, N., Tao, D., Chen, Y., & Zhang, M. One For All: Towards Training One Graph Model For All Classification Tasks. In The Twelfth International Conference on Learning Representations.
>
> [2] Wang, Z., Zhang, Z., Chawla, N., Zhang, C., & Ye, Y. (2024). Gft: Graph foundation model with transferable tree vocabulary. Advances in Neural Information Processing Systems, 37, 107403-107443.

---

> > ### Comment · Reviewer_PsLJ · 2025-08-04
> >
> > Thanks for the reply. I read through the replies carefully and I overviewed other reviewers' suggestion as well. The authors have addressed the concerns of the baseline selection.
> >
> > However, the idea of domain prototype and the semantic anchor are not initially proposed, though the authors argue that it is new to conduct them in this setting. Also, I am concerned about the privacy leakage of  local domain prototype sharing with the server.
> >
> > I would like to raise my score to 3.

---

> ### Author Response · Authors · 2025-08-05
> **Further response**
>
> Thank you for your follow-up and the score adjustment.. We agree that the privacy implications of prototype sharing merit deeper theoretical analysis, and will evaluate these privacy risks in future work.

---

### Official Review · Reviewer_v6fQ · 2025-07-01

**Clarity:** 3
**Significance:** 3
**Originality:** 3
**Rating:** 5
**Confidence:** 4

**Summary:**

This paper delves into the Federated Graph Foundation Model (FedGFM), which seamlessly integrates two areas within graph machine learning: Federated Graph Learning (FGL) and Graph Foundation Models (GFM). The paper offers a practical overview of the challenges associated with both FGL and GFM in real-world applications, while emphasizing their complementary strengths. FedGFM is introduced as a novel framework for building graph foundation models by harnessing distributed datasets from multiple parties and domains, along with decentralized computational resources. A central issue identified in this context is knowledge entanglement, where global models struggle to separate domain-specific insights into distinct representations. To tackle this, the authors propose FedGFM+, which incorporates the AncDAI and AdaDPP modules. Comprehensive experiments across various datasets validate the practicality and performance of FedGFM+.

**Questions:**

See weakness

**Ethical Concerns:**

["NO or VERY MINOR ethics concerns only"]

**Final Justification:**

Most of my concerns have solved

**Limitations:**

See weakness

**Quality:**

3

**Strengths And Weaknesses:**

Strengths:

1. This work presents FedGFM, a novel framework combining Federated Graph Learning (FGL) with Graph Foundation Models (GFM). It leverages multi-domain data and distributed resources, marking the first study in this area.

2. FedGFM+ builds on FedGFM, addressing communication challenges and knowledge entanglement with robust empirical methods.

3. Experiments on eight datasets across tasks validate its effectiveness, with baselines including GNNs, FGL variants, and federated GFM adaptations.

Weakness:

1. In the context of VQ-VAE or Graph-VQ-VAE, the concept of vocabulary collapse [1, 2] refers to a situation where only a small subset of tokens in the codebook is frequently accessed, while many are rarely or never used. How does this differ from the knowledge entanglement issue discussed in this paper? Please elaborate on the distinction.

2.  What is the reason for structuring the FedGFM paradigm’s pre-training phase as iterative rounds of federated communication, while the fine-tuning phase is carried out independently by each client?

3. Provide further explanation regarding the fairness of comparisons made between the FGL baselines and FedGFM+.

4. Could you clarify the meaning and implications of Figure 2 in more detail?

[1] Zheng, C., & Vedaldi, A. (2023). Online clustered codebook. In Proceedings of the IEEE/CVF International Conference on Computer Vision (pp. 22798-22807).
[2] Wang, Z., Zhang, Z., Chawla, N., Zhang, C., & Ye, Y. (2024). Gft: Graph foundation model with transferable tree vocabulary. Advances in Neural Information Processing Systems, 37, 107403-107443.

---

> ### Author Rebuttal · Authors · 2025-07-28
>
> We sincerely thank you for your constructive comments, and initially address all of your concerns below:
>
> ---
>
> **Question 1**: What is the difference between knowledge entanglement in FedGFM and vocabulary collapse in VQ-VAE?
>
> We appreciate the opportunity to clarify the distinction between the two challenges.
>
> - ***Knowledge Entanglement*** is a unique challenge in FedGFM caused by the lack of centralized optimization over heterogeneous, domain-specific data. This leads to different semantics being mapped to the same token positions, resulting in conflicting updates during server-side aggregation. For example, token ID 1 may represent financial concepts on one client and biomedical knowledge on another. These conflicts degrade the quality of the aggregated global model and cause domain confusion downstream.
>
> - ***Vocabulary Collapse*** is a common issue in VQ-VAE, where only a few codebook tokens dominate usage while others remain inactive. This self-reinforcing dynamic occurs because tokens slightly closer to inputs are selected and updated more often, further increasing their dominance. Consequently, token utilization decreases and representational diversity suffers.
>
>
> ---
>
> **Question 2**: Why is the federation mechanism not applied during the fine-tuning stage?
>
> Thank you for your comment. FedGFM consists of two stages. During pre-training, clients act as data holders, collaboratively training a unified and generalizable graph foundation model via federated communication over distributed, domain-specific datasets. In contrast, during fine-tuning, clients serve as model users or task adapters, working independently without federation. Notably, the fine-tuning data often differs from the pre-training data, reflecting practical deployment scenarios where distributed coordination is **unnecessary or impractical**. Based on the above considerations, federated communication is employed during the pre-training stage, but not during the fine-tuning stage.
>
>
> ---
>
> **Question 3**: Explain the fairness of the comparison between FGL baselines and FedGFM+
>
>
> Thank you for the question. We appreciate the opportunity to clarify the fairness of our comparison.
>
> All methods (including FedGFM+ and the FGL baselines) are evaluated under the same federated protocol, with consistent feature pre-processing, identical data splits, and shared optimization settings. The only difference lies in the modeling paradigm: FedGFM+ adopts a federated pre-training and fine-tuning framework, while FGL baselines follow a task-specific federated training approach. This paradigm shift allows FedGFM+ to enable generalized collaboration across domains and tasks (see Lines 64–78), whereas FGL baselines are inherently limited to single-task, task-bound collaboration (see Lines 38–50).
>
> **In summary**, the performance **superiority of FedGFM+ arises from its modeling innovations**, not from any unfair advantage in experimental setup. We therefore consider the comparison between FedGFM+ and FGL baselines to be **fair and meaningful**.
>
> ---
>
> **Question 4**: Clarify the meaning of Figure 2 in more detail.
>
> Figure 2 is designed to illustrate the core challenge of the FedGFM paradigm, named knowledge entanglement.
>
> Specifically, we consider three datasets from distinct domains and tasks: Cora (citation network, node classification), WN18RR (knowledge graph, edge classification), and HIV (molecular graph, graph classification). As shown in Figure 2(a), these datasets exhibit significantly different structural patterns, while Figure 2(b) ('Raw Feat.') shows notable feature disparities across domains.
>
> We then compare GFT and its naive federated variant GFT\*:
>
> - **GFT**, trained centrally across all datasets (jointly optimized);
>
> - **GFT\***, trained in a decentralized manner following the FedGFM setting.
>
> As shown in Figure 2(b), the embeddings produced by GFT ('GFT Emb.') preserve clear domain-specific separability, while those produced by GFT\* ('GFT\* Emb.') collapse into indistinguishable representations.
>
> This result highlights a fundamental limitation in the FedGFM setting: knowledge entanglement, where multi-domain knowledge is encoded into indistinguishable representations, thereby limiting downstream adaptation.
>
> ---

---

> > ### Comment · Reviewer_v6fQ · 2025-08-05
> >
> > Thank you for the detailed response.  The author has answered most of my questions, but I still have some concerns.
> >
> > **Regarding Q2 (Why is the federation mechanism not applied during the fine-tuning stage?)**, there are existing studies on federated fine-tuning of pre-trained base models that show the benefits of collaboration during fine-tuning. The authors should further clarify how this work differs from these federated fine-tuning methods.
> >
> > **Regarding Q3 (Explain the fairness of the comparison between FGL baselines and FedGFM+)**, since the authors mentioned that FGL baselines apply only to a limited set of tasks, it would be helpful to explain, for each method, why it is not applicable to certain tasks.
> >
> > Reference: [1] Wang, Z., Shen, Z., He, Y., Sun, G., Wang, H., Lyu, L., & Li, A. (2024). Flora: Federated fine-tuning large language models with heterogeneous low-rank adaptations. Advances in Neural Information Processing Systems, 37, 22513-22533.

---

> ### Author Response · Authors · 2025-08-05
> **Further response**
>
> Thank you for your follow-up. We are glad our previous response addressed most of your concerns. Below we provide further clarifications on the remaining points.
>
> ---
>
> **Question 2**: Clarify how this work differs from these federated fine-tuning methods.
> We appreciate your reference to existing work on federated fine-tuning. Here we clarify how the FedGFM paradigm differs from these approaches, and why this motivates our choice to execute the fine-tuning phase in isolation rather than in a federated manner.
>
> In existing federated fine-tuning methods for foundation models, the primary goal is to adapt the model to multiple local datasets held by clients; in such settings, the fine-tuning data directly corresponds to the clients’ local data.
>
> By contrast, in FedGFM, multiple decentralized datasets from clients are used only in the pre-training stage to learn a GFM. In the fine-tuning stage, the learned GFM is adapted to a target graph that may originate outside of any participating clients. This naturally allows (and in some scenarios necessitates) performing fine-tuning in isolation rather than through a federated process.
>
> ---
>
> **Question 3**:  Explain the reason for each FGL baseline does not apply to certain tasks.
>
> Thank you for your suggestion. Below, we provide a more detailed explanation of why existing federated graph learning methods are not directly applicable to our setting:
>
> - FedSage+, Fed-PUB: Designed for subgraph-level federated learning; their model architectures and optimization objectives are not suitable for graph classification datasets.
>
> - FedGTA, FedTAD, FGSSL, FGGP: Developed for subgraph-level federated learning, but their algorithms rely on node label inputs, making them applicable only to node classification tasks—not to edge classification datasets (which lack node labels).
>
> - GCFL+, FedStar: Designed for graph-level federated learning; their model architectures and optimization objectives are not suitable for subgraph-level federated learning with node or edge classification tasks.
>
> ---
>
> We hope this resolves the remaining ambiguities. Thank you again for your thoughtful feedback and for recognizing our contributions.

---

> > ### Author Response · Authors · 2025-08-07
> > **Gentle Inquiry on Remaining Questions**
> >
> > Dear Reviewer,
> >
> > As the discussion period is nearing its end, may we kindly ask if you have any further concerns or points requiring clarification? Thank you again for your time and effort in reviewing our work.
> >
> > Sincerely,
> >
> > Authors of Submission 11756

---

> > ### Comment · Reviewer_v6fQ · 2025-08-07
> >
> > Thanks for the responses, most of my concerns have resolved, I decide to raise my score.

---

### Official Review · Reviewer_cFwZ · 2025-07-01

**Clarity:** 4
**Significance:** 3
**Originality:** 3
**Rating:** 5
**Confidence:** 4

**Summary:**

This paper investigates Federated Graph Foundation Models (FedGFM) and proposes a unified training framework, FedGFM+, that aims to combine the complementary advantages of Federated Graph Learning (FGL) and Graph Foundation Models (GFM) and mitigate their respective limitations. The authors identify and discuss key limitations in both areas and introduce two novel components (AncDAI and AdaDPP) to mitigate the issue of knowledge entanglement across domains. The experimental evaluation covers eight graph datasets, multiple downstream tasks, and diverse baselines, demonstrating the consistent superiority of FedGFM+.

**Questions:**

Please refer to the weaknesses.

**Ethical Concerns:**

["NO or VERY MINOR ethics concerns only"]

**Final Justification:**

Most of my initial concerns have been addressed properly during the rebuttal. Given the core contributions of this work. I will raise my score to 5.

**Limitations:**

Yes.

**Quality:**

3

**Strengths And Weaknesses:**

### Strengths:
- This paper presents a novel and practically relevant study grounded in real-world scenarios about Federated Graph Foundation Models (FedGFM), proposing a training paradigm that unifies decentralized graph data sources with the goal of learning a generalizable graph foundation model.
- The paper provides a diagnosis of core limitations in existing Federated Graph Learning (FGL) and Graph Foundation Model (GFM) approaches. Based on these insights, the authors further proposed the FedGFM+ framework, which organically combines FGL and GFM and complementarily addresses the limitations of both.
- The authors provide an empirical analysis that evaluates FedGFM from the perspective of feasibility, effectiveness, and further experimentally reveal the non-trivial challenge faced by simple federated variants of the naive centralized GFM training strategy, namely, knowledge entanglement.
- The experiments are comprehensive, and the performance is promising.
- Overall, the manuscript is well-structured with well-designed illustration figures and tables, and clear technical explanations.
- The authors have provided the source code of this paper.

### Weaknesses:
- It would be beneficial for the authors to further clarify the workings of the proposed FedGFM+ framework by providing detailed algorithmic pseudo code. A more formal description would aid readers who are less familiar with FGL or GFM in fully understanding the methodology.
- In the empirical analysis section, the experiment on the knowledge entanglement problem measured the encoding differences of a simple federated variant of the centralized GFM training strategy GFT for graph datasets in different domains; I suggest that the authors can supplement the encoding results of FedGFM+ to better illustrate that FedGFM+ can indeed alleviate the knowledge entanglement problem.
- The author divides different datasets into 3 clients. I suggest that the author further explore the performance of tasks divided into other numbers of clients. For example, does dividing Cora into 5 clients affect the performance of the learned graph-based model?

---

> ### Author Rebuttal · Authors · 2025-07-28
>
> We sincerely thank you for your constructive comments, and we will initially address all of your concerns below:
>
> ---
>
> **Question 1**: Lack of formal algorithmic pseudo code for FedGFM+
>
> We fully agree with this constructive feedback and apologize for the oversight. Here, we present the pseudo code for the federated pre-training phase and isolated fine-tuning phase of FedGFM+ in Algorithms 1 and 2, respectively. We are committed to revising this content to provide a more rigorous and formal exposition of the FedGFM+ approach in the revision stage.
>
>
> **Algorithm 1 - Federated Pre-Training Phase of FedGFM+**
>
> **Input**: Number of clients $K$, each client's local graph $G^k$, initialized global gVQ-VAE parameters $\Theta^{g}$, each client's local learnable prompt parameter $\Phi^k$ and projection vector $w^k$, number of communication rounds $T$, number of local epochs $E$, noise scaling factor $\sigma$.
>
>
>
> **Output**: learned global GFM $\Theta^{g}$, domain-aware prompt pool $\rho$ and projection pool $w$
>
>
>
> ***for*** each client $k$ in $1,...,K$ ***do***
>
> ------------ Initialize local gVQ-VAE with global gVQ-VAE, $\Theta^k \leftarrow \Theta^g$.
>
> ------------ compute domain prototype $p^k$ via Eqs. (3), (4)
>
> ------------ upload domain prototype to central server
>
> ***end***
>
>
>
> compute perturbed embeddings and initialize global codebook via Eq. (5)
>
>
>
> ***for*** each round $t$ ***in*** $1,...,T$ ***do***
>
> ------------ ***for*** each client $k$ in $1,...,K$ ***do***
>
> ------------------------- replace local gVQ-VAE with global gVQ-VAE, $\Theta^k_t \leftarrow \Theta^g_t$
>
> ------------------------- ***for*** local epoch $e$ in $1,...,E$ ***do***
>
> ------------------------------------- local graph feature prompting via Eq. (6)
>
> ------------------------------------- local pre-training to update local gVQ-VAE $\Theta^k_t$ and learnable prompt $\Phi^k$, $w^k$ via Eq. (2)
>
> ------------------------- ***end***
>
> ------------------------- upload gVQ-VAE parameters $\Theta^k_t$ to central server
>
> ------------ ***end***
>
> ------------ aggregate local  gVQ-VAE parameters to update global  gVQ-VAE, $\Theta^g_{t+1} \leftarrow \frac{N_k}{N}\sum_{k=1}^{K}\Theta^k_t$
>
> ***end***
>
>
> ---
>
>
> **Algorithm 2 - Isolated Fine-Tuning Phase of FedGFM+**
>
> **Input**: Target graph $G$, learned global GFM $\Theta^{g}$, domain-aware prompt pool $\rho$ and projection pool $w$, number of fine-tuning epochs $E^t$, task head $W$.
>
>
>
> **Output**: fine-tuned global GFM $\Theta^{g}$ and task head $W$.
>
> ***for*** each epoch $e$ ***in*** $1,...,E^t$ ***do***
>
> ------------ target graph feature prompting via Eq. (7)
>
> ------------ downstream task-specific optimization to fine-tune global GFM $\Theta^{g}$ and $W$.
>
> ***end***
>
>
>
>
> ---
>
> **Question 2**: Missing encoding results to demonstrate how FedGFM+ alleviates the knowledge entanglement problem.
>
> Thank you for your valuable feedback. In Section 3, we show the similarity of initial node features and GFT/GFT* embeddings for Cora, WN18RR, and HIV, which highlights the knowledge entanglement issue in FedGFM—where representations from different domains become indistinguishable. As suggested, we further report FedGFM+ embeddings on these datasets. The results are summarized in Table. 1:
>
> **Table. 1: Comparison of the distinguishability of FedGFM+ and baseline embeddings on cross-domain datasets.**
> |                | Cora-WN18RR | Cora-HIV | WN18RR-HIV |
> | -------------- | ----------- | -------- | ---------- |
> | **Raw Feat.**      | 0.53        | 0.41     | 0.49       |
> | **GFT**            | 0.72        | 0.64     | 0.71       |
> | **GFT***           | 0.98        | 0.92     | 0.96       |
> | **FedGFM+ (Ours)** | 0.66        | 0.52     | 0.65       |
>
> As observed, FedGFM+ produces significantly lower embedding similarity across the three domains compared to GFT and GFT*, indicating that its AncDAI and AdaDPP modules effectively help encode multi-domain semantics into distinguishable representations, thereby mitigating the knowledge entanglement issue.
>
> ---
>
> **Question 3**: How does FedGFM+ perform under increased number of clients?
>
> We appreciate the insightful suggestion. Following your advice, we evaluated FedGFM+ and 5 representative baselines, including FedAvg, Fed-PUB, GCFL+, OFA\*, and GFT\* on the Cora, PubMed, WikiCS, OGB-arxiv, FB15K237, WN18RR, HIV and PCBA datasets, each evaluated under both 3-client and 5-client partitions. The results are summarized in **Table. 2**.
>
>  **Table. 2: Performance comparison between FedGFM+ and baselines under 5 client partitions.**
>
> |         | Cora  | PubMed | OGB-arxiv | WikiCS | FB15K237 | WB18RR | HIV   | PCBA  |
> | ------- | ----- | ------ | --------- | ------ | -------- | ------ | ----- | ----- |
> | FedAvg  | 67.42 | 70.45  | 66.30     | 64.52  | 65.28    | 65.39  | 60.42 | 63.55 |
> | Fed-PUB | 69.20 | 73.18  | 67.22     | 65.37  | 67.50    | 66.20  | N/A   | N/A   |
> | GCFL+   | N/A   | N/A    | N/A       | N/A    | N/A      | N/A    | 60.49 | 63.83 |
> | OFA*    | 68.42 | 72.35  | 67.40     | 63.28  | 67.25    | 64.82  | 59.45 | 63.81 |
> | GFT\*   | 69.88 | 73.52  | 66.59     | 66.73  | 68.53    | 65.38  | 58.31 | 64.24 |
> | FedGFM+ | 72.42 | 76.12  | 70.24     | 69.24  | 71.42    | 68.40  | 62.18 | 66.04 |
>
> As observed, FedGFM+ **consistently outperforms** all baselines. Compared to the 3-client setting in Section 3, all methods show performance drops under 5-client partitioning, which is a common phenomenon in federated learning when data becomes more sparsely distributed (increased client heterogeneity).
>
>
> ---

---

> > ### Comment · Reviewer_cFwZ · 2025-08-03
> >
> > I would like to thank the authors for their detailed rebuttal. Most of my initial concerns have been addressed, and I appreciate the clarifications.
> >
> > Notably, one minor issue remains: the supplemented pseudocode for FedGFM+ is still lacking in some critical implementation details (particularly regarding the fine-tuning phase. It would be helpful if the authors could explicitly clarify what loss functions are used for both the task head and the fine-tuning of the GFM).
> >
> > Despite this, I believe the core contributions are novel and well-motivated, and the paper is generally well-presented. Therefore, I will raise my score to 5.

---

> > > ### Author Response · Authors · 2025-08-04
> > > **Further response**
> > >
> > > Thank you for your follow-up. We’re glad our previous response addressed most of your concerns. Regarding the pseudocode details, we appreciate your attention and provide clarification below:
> > >
> > > ---
> > >
> > > During federated pretraining, FedGFM+ learns a cross-domain GFM with parameters $\Theta^g$, along with a domain-aware prompt pool $\rho$ and a projection pool $w$.
> > >
> > > In the fine-tuning phase for downstream tasks (e.g., graph, edge, or node classification), we proceed as follows:
> > >
> > > Input the target graph into the pretrained modules to obtain node embeddings.
> > >
> > > - For graph classification, average all node embeddings to get a graph-level representation.
> > >
> > > - For edge classification, average the embeddings of node pairs to get edge-level representations.
> > >
> > > - For node classification, use node embeddings directly without pooling.
> > >
> > > These representations are then passed to a linear classifier for label prediction, and fine-tuning is performed using cross-entropy loss.
> > >
> > > ---
> > >
> > > We hope this clarifies the remaining ambiguity. Thank you again for your thoughtful feedback and your acknowledgement of our contributions.

---

### Official Review · Reviewer_GZh8 · 2025-07-02

**Clarity:** 3
**Significance:** 3
**Originality:** 4
**Rating:** 4
**Confidence:** 4

**Summary:**

The paper introduces FedGFM, a federated learning framework for training graph foundation models (GFMs). A key challenge identified is knowledge entanglement, where representations from different domains become indistinguishable, limiting downstream performance. To address this, the authors propose FedGFM+, which extends FedGFM with two modules: (1) AncDAI, a global anchor-based domain-aware initialization that encodes domain-specific prototypes before pretraining to ensure separable representations across domains; and (2) AdaDPP, an adaptive domain-sensitive prompt pool used during fine-tuning to enhance domain-specific adaptation via learned prompts from each client. The authors support their method with theoretical analysis and demonstrate improved performance across multiple datasets compared to strong baselines, including domain-specific and federated models.

**Questions:**

1.	How is the model trained to ensure well-separated prototypes across domains?
2.	Can you provide a pseudo-code or algorithm sketch for FedGFM+?
3.	Is it possible to evaluate or estimate the privacy leakage of the method?
4.	How does the model perform on domains with limited or imbalanced data?
5.	What happens to performance if the data is randomly partitioned or if the number of clients increases?

**Ethical Concerns:**

["NO or VERY MINOR ethics concerns only"]

**Final Justification:**

After carefully considering the rebuttal and follow-up clarifications, as well as the discussions, I summarize my assessment as follows:

**Resolved Issues:**

- The authors added explicit details on the computation and communication costs (48 MB per client per round, ~9.6 s local execution time).
- They clarified the applicability of the framework to textual-attributed graphs. However, they acknowledged the limitation for more general heterogeneous cases.
- They explained their partitioning strategy, noting Louvain for subgraph-level FL and uniform random for graph-level datasets.  Although I still believe random partitioning is an important challenge for subgraph-FL problems.

**Unresolved Issues:**

- Privacy risks: Although the authors now list the exchanged information and potential risks, no quantitative analysis or mitigation strategies are provided. In a federated learning setting, this remains a central limitation.
- Scalability/robustness: While communication numbers are given, there is no analysis of how these costs scale with more clients, larger models, or longer training. Random partitioning robustness is still not fully explored.

**Weighting:**

- The paper’s motivation, novelty, and strong empirical results remain its key strengths.
- The unresolved privacy and robustness aspects are important, but do not fully undermine the technical contribution.
- Overall, the paper makes a valuable step in defining and addressing the FedGFM paradigm, and I maintain my positive borderline-accept score.

**Limitations:**

1.	The scalability of the method with respect to the number of domains is not discussed.
2.	The potential privacy leakage due to sharing prototypes and prompts is not analyzed.

**Paper Formatting Concerns:**

No Formatting. Concerns.

**Quality:**

3

**Strengths And Weaknesses:**

**Strengths:**
	1.	The idea of applying federated learning to train graph foundation models is important and addresses a fundamental need in graph machine learning.
	2.	The paper identifies a key challenge in federated graph foundation models—knowledge entanglement—and supports this with a clear and relevant experimental analysis.
	3.	The proposed solutions are backed by both theoretical analysis and strong empirical results.
	4.	The overall structure and flow of the paper are logical and easy to follow.

**Weaknesses:**
	1.	Although Theorem B.1 shows that prototypes are distinguishable even under random initialization, the paper does not provide any quantitative estimation of how separated they are. Including such an estimate, perhaps with a simple example, would help illustrate this point more clearly.
	2.	A similar concern applies to Theorem B.2. While the theorem suggests that the codebook is more likely to assign distinct tokens to different domains, it’s unclear how likely this is in practice. Quantitative or empirical evidence would strengthen the argument.
	3.	Theorem B.2 is described as general, but the proof only focuses on cosine similarity. It would be helpful to clarify whether the conclusions hold under other similarity metrics.
	4.	Since all domain prototypes and prompts (from Sections 4.1 and 4.2) are shared with the server, there should at least be a discussion of the potential privacy risks associated with this sharing.
	5.	Based on Equation (7), all prompts appear to have the same dimensionality as the node features, which implies that all node features across graphs must have the same dimension. This is a strong assumption and could limit the applicability of the method to heterogeneous graphs.
	6.	Because each prompt has the same dimension as the node features and requires its own projection weights, the resulting objective function becomes high-dimensional and potentially expensive to optimize. This can increase both computation and communication costs, especially in a decentralized setting. More justification and analysis of this overhead would be beneficial.
	7.	While the experimental results are thorough and span multiple tasks and domains, the paper does not report any comparison of computation or communication costs.
	8.	Introducing synthetic prototypes to the codebook increases its size, thereby enlarging the search space for the optimization in Equation (1). This added complexity should be discussed.

---

> ### Author Rebuttal · Authors · 2025-07-28
>
> We sincerely thank you for your comments, and appreciate that you acknowledged the strengths of our work. Here we initially address all concerns below:
>
> ---
>
>
>
> **Question 1**: Theorem B.1 and B.2 lack quantitative estimation or empirical evidence.
>
> Thank you for your insightful feedback. We hope to address your concerns.
>
> For Theorem B.1 (Domain Prototype Distinguishability), prior work has shown that even randomly initialized encoders can produce distinguishable embeddings for multi-domain data, especially in the text domain [1]. This supports our assumption that distinguishability can emerge from structural biases and feature heterogeneity.
>
> For Theorem B.2 (Semantic Separability of the AncDAI-Initialized Codebook), we have added empirical evidence demonstrating that FedGFM+ encodes multi-domain knowledge into semantically separable codebook representations. Please refer to our response to Question 2 from reviewer cFwZ.
>
> ---
>
>
>
> **Question 2**: Whether the theorem B.2 hold under other similarity metrics?
>
> Thank you for your comments. Theorem B.2 remains valid for other similarity metrics (e.g., Euclidean distance), as long as the codebook’s query strategy is adapted accordingly. Since the codebook selects discrete tokens based on encoder outputs, the similarity metric used in the query must match the one assumed in the theorem. In our case, Theorem B.2 is based on cosine similarity, which is also the metric used in our codebook querying.
>
>
>
> ---
>
> **Question 3**: Lack of discussion about the potential privacy risks associated with shared domain prototypes and prompts.
>
> We sincerely thank you for your comments and agree that a deeper privacy risk analysis of the shared domain prototypes and prompts in FedGFM+ would provide stronger safeguards. However, we respectfully emphasize that we did not upload any raw local data; instead, it was encoded using a randomly initialized model, which helps reduce the risk of privacy leakage to some extent. It is also important to note that privacy preservation is not the primary focus of this work. Our main contribution lies in defining a new paradigm named federated graph foundation model (FedGFM). In future revisions, we plan to include a theoretical analysis and explore the use of noise injection on the uploaded domain prototypes to enhance privacy protection.
>
>
>
> ---
>
> **Question 4**: Eq. (7) results in same feature dimension for  all nodes, which may limit the applicability of the method to heterogeneous graphs.
>
>
>
> Thank you for your comments. Equation 7 illustrates the feature augmentation process when applying the pre-trained GFM and personalized prompt pool to a specific target graph during fine-tuning. We would like to clarify that the statement "all nodes must have the same feature dimension" does not imply that our method is unsuitable for heterogeneous graphs. On the contrary, our approach is applicable to all textual attribute graphs with textual descriptions for nodes and edges. We ensure a consistent feature dimension by encoding all textual descriptions into unified vector representations using a shared pre-trained language model (e.g., Sentence-BERT, as used in this paper).
>
>
>
> ---
>
> **Question 5**: Need more justification and analysis of computation and communication costs about  would prompts and projections.
>
> Thank you for your comment. Our prompts are very lightweight and contain far fewer parameters than the backbone network. Although each prompt shares the same feature dimension as the node features, the total number of prompt parameters remains minimal, resulting in negligible additional computation and communication overhead in the federated setting. Specifically, in our experiments, for each client, we use only 3 learnable prompts (shape: 3 × 768) along with a projection matrix (shape: 768 × 3), which is extremely lightweight.
>
>
>
> ---
>
>
>
> **Question 6**: How is the model trained to ensure well-separated prototypes across domains?
>
> Thank you for your comments. We respectfully clarify that FedGFM+ relies on the AncDAI module to yield distinguishable domain prototypes for graphs from different domains. As outlined in Section 4.1 and Theorem B.1, prototype computation is independent of a pre-trained encoder; it only presupposes that every client employs an identically randomly initialized encoder.
>
>
>
> ---
>
>
>
> **Question 7**: provide a pseudo-code or algorithm sketch for FedGFM+.
>
> We sincerely appreciate your constructive comments. Due to word limit, please see our rebuttal to **Question 1** from reviewer **cFwZ**.
>
>
>
> ---
>
>
>
> **Question 8**: How does the model perform on domains with limited or imbalanced data?
>
>
>
> Thank you for your constructive feedback. We respectfully remind you that the original manuscript briefly discussed the performance of FedGFM+ on limited data and domain imbalanced data. Specifically:
>
> - ***Limited Data***: In Appendix D of the original manuscript, we evaluated the performance of FedGFM+ in the few-shot setting. As observed, FedGFM+ consistently outperforms naive federated adaptations of centralized GFM training strategies across all evaluated settings
> - ***Imbalanced Data***: We’d like to clarify that our original experiments were conducted on data with imbalanced domains (see Appendix C.1 Table 4 and C.2 Table 5). The graph classification dataset involves significantly more nodes than the node classification dataset, based on local codebook access counts. To address this, we use a fixed batch size and number of batches with random sampling during pre-training, ensuring each client sees similar improvement per training round. Our results show that FedGFM+ handles domain imbalance effectively. We will include a clearer explanation of this training strategy in the revised version.
>
>
>
> ---
>
> **Question 9**: What happens to performance if the data is randomly partitioned or if the number of clients increases?
>
> We sincerely appreciate your constructive comments. Due to word limit, please see our rebuttal to **Question 3** from reviewer **cFwZ**.

---

> > ### Author Response · Authors · 2025-08-01
> > **Additional References**
> >
> > Dear Reviewer,
> >
> >
> > We'd like to add a reference supporting **Question 1**. Apologies for the omission. Thank you for your time.
> >
> >
> > [1] *Wieting, J., & Kiela, D. (2019). No training required: Exploring random encoders for sentence classification. arXiv preprint arXiv:1901.10444.*

---

> > ### Comment · Reviewer_GZh8 · 2025-08-04
> >
> > Thank you to the authors for the detailed rebuttal. I appreciate the effort to respond to many points despite the word limit. However, I believe that most of my original concerns remain insufficiently addressed. In particular:
> > 1. Privacy Considerations: While the authors note that the use of randomly initialized encoders reduces leakage risk, the paper still lacks a proper discussion of what information is exchanged (e.g., prototypes and prompts) and what privacy vulnerabilities this might introduce. Even a brief qualitative discussion or threat model would strengthen the work. This is the most significant unresolved issue, especially for a federated learning setting.
> > 2. Applicability to Heterogeneous Graphs: The rebuttal clarifies that the method relies on textual embeddings (e.g., Sentence-BERT) to ensure feature consistency. While this is a valid strategy, it imposes a non-trivial assumption that should be made explicit in the paper, as it limits the framework’s applicability to more general (non-textual or structurally heterogeneous) graph modalities.
> > 3. Computation and Communication Overhead: The response states that prompt parameters are lightweight, which is appreciated. However, a clearer analysis—perhaps including actual memory/communication cost per client and computation time for each training step—would improve transparency and help readers assess the method’s feasibility in real deployments.
> > 4. Performance under random partitioning is critical to understanding robustness. Also, weakness 8 is not adressed.
> >
> > Overall, the paper tackles an important and novel problem with a promising approach. The contributions are clear, and the empirical results are strong. However, the lack of privacy discussion and missing evaluations related to scalability and overhead prevent a higher score. I maintain my original rating.

---

> > > ### Author Response · Authors · 2025-08-05
> > > **Further response**
> > >
> > > Thank you for your follow-up. We are pleased that you appreciate the motivational contributions and experimental performance of our work. We will now address your remaining concerns:
> > >
> > >
> > >
> > > ---
> > >
> > >
> > >
> > > **Question 1: (Privacy Considerations) Lacks a proper discussion of what information is exchanged (e.g., prototypes and prompts) and what privacy vulnerabilities this might introduce.**
> > >
> > > We appreciate this important question regarding privacy protections in our framework. Below, we provide a discussion of the information exchange and associated privacy considerations:
> > >
> > > - **Information Exchange in FedGFM+**: During federated pretraining, clients upload: Domain prototypes $p$ computed from local subgraphs with randomly initialized encoder (Eq. 4); Complete graph foundation model parameters ($\Theta$) including: Encoder, Codebook, Decoder; Domain-aware prompts ($\phi$) and projection ($w$) after pre-training. Server broadcast: graph foundation model parameters ($\Theta^g$).
> > >
> > > - **Threat Model**: We assume an honest-but-curious server, which follows the protocol but may attempt to infer sensitive client data from the received information. Similarly, a small fraction of clients could be malicious and try to reconstruct other clients’ private data.
> > >
> > > - **Potential Privacy Risks** : (1) Model inversion: The server could attempt to approximate client-specific feature patterns from the model parameters (GFM and learnable prompts); (2) Community Inference: The server may infer the community structure across clients based on the similarity between pairwise clients' domain prototypes; (3) Server-side Feature reconstruction: The server may use each client's uploaded domain prototype to infer the original features; (4) Client-side Feature reconstruction: After receiving the global vocabulary initialized by AncDAI, clients may infer the original data of other clients through the domain prototypes in it.
> > >
> > >
> > >
> > > ---
> > >
> > >
> > >
> > > **Question 2: (Applicability to Heterogeneous Graphs) The framework is Limited by applicability to more general (non-textual or structurally heterogeneous) graph modalities.**
> > >
> > >
> > >
> > > Thank you for your constructive feedback. We acknowledge that our current implementation of the framework is not applicable to non-text attribute graphs and other structurally heterogeneous graphs. However, we respectfully remind you that for non-text attribute graphs, the feature space can be aligned to make our framework applicable via [1]. However, we understand that we have not yet added specific experiments to evaluate the performance, which will be a focus of our future work.
> > >
> > > ---
> > >
> > > **Question 3: (Computation and Communication Overhead) Needs a clearer analysis about actual memory/communication cost per client and computation time for each training step).**
> > >
> > > Thank you for your constructive feedback. Based on the reviewer's suggestions, we have added the actual local running time of each client during the pre-training phase, as well as the average communication cost per client per round:
> > >
> > > - **Average communication cost (per client):** 48 MB. This amount of communication should be evaluated in the context of our scenario (training the base model in a federated environment) and is therefore acceptable.
> > > - **Maximum actual local execution time:** 9.58 seconds, which means that even without an asynchronous federated learning strategy, all clients can transmit their local information to the server in no more than 9.58 seconds.
> > >
> > > ---
> > >
> > > **Question 4: (Robustness) Needs Performance under random partitioning for understanding robustness.**
> > >
> > > We respectfully note that the original partitioning strategy follows a widely adopted FGL framework and is not inherently less suited for evaluating robustness in real-world scenarios than random partitioning.
> > >
> > > - **Node and edge classification datasets** belong to the subgraph-level federated learning setting. We use the Louvain algorithm (the most common subgraph partitioning strategy [2]), which produces subgraphs with well-defined communities by maximizing modularity. In contrast, simple random partitioning (e.g., by node or edge labels) often yields unrealistic subgraphs, such as those with weak connectivity or extreme sparsity.
> > >
> > > - **Graph classification datasets** belong to the graph-level federated learning setting. Here, we employ uniform random partitioning, assigning each graph to a client with equal probability. This choice is motivated by the fact that the graph classification datasets we use consist of multiple parallel classification tasks.
> > >
> > > ---
> > >
> > > **References**
> > >
> > > [1] *Zhao, H., Chen, A., Sun, X., Cheng, H., & Li, J. (2024, August). All in one and one for all: A simple yet effective method towards cross-domain graph pretraining. In Proceedings of the 30th ACM SIGKDD Conference on Knowledge Discovery and Data Mining (pp. 4443-4454)*
> > >
> > > [2] *Li, X., Zhu, Y., Pang, B., Yan, G., Yan, Y., Li, Z., ... & Wang, G. (2024). Openfgl: A comprehensive benchmark for federated graph learning. arXiv preprint arXiv:2408.16288.*

---

> > > > ### Author Response · Authors · 2025-08-07
> > > > **Gentle Inquiry on Remaining Questions**
> > > >
> > > > Dear Reviewer,
> > > >
> > > > As the discussion period is nearing its end, may we kindly ask if you have any further concerns or points requiring clarification?  Thank you again for your time and effort in reviewing our work.
> > > >
> > > > Sincerely,
> > > >
> > > > Authors of Submission 11756

---

> > > > > ### Comment · Reviewer_GZh8 · 2025-08-08
> > > > > **Final Response**
> > > > >
> > > > > I thank the authors for the added clarifications and detailed follow-up. These additions, especially the explicit description of the information exchange, the potential privacy risks, and the computation/communication statistics, can improve the quality and transparency of the paper.
> > > > >
> > > > > That said, given that there are still unresolved privacy risks inherent to the framework and no quantitative privacy leakage analysis or mitigation strategies presented, I believe this remains an important limitation. I will therefore maintain my current positive score.

---

### Decision · Program_Chairs · 2025-09-17

**Decision:**

Accept (poster)

**Comment:**

After carefully reading the submission, the reviewers' comments, and the authors' detailed rebuttal, I recommend acceptance.

The paper presents FedGFM, a novel paradigm for building GFMs by leveraging decentralized data and computational resources from multiple clients. The authors identify a core challenge in this setting, which they term "knowledge entanglement," where a globally aggregated model struggles to maintain domain-specific representations, thereby hampering downstream performance on diverse local tasks. To address this, the paper proposes an enhanced framework, which introduces two novel components, AncDAI and AdaDPP. These are designed to ensure that representations from different clients remain distinguishable during the pretraining stage and to help the model adapt more effectively to local downstream tasks during fine-tuning. Many reviewers acknowledged the novelty of the problem formulation and the proposed method. The submission is backed by a comprehensive and convincing experimental evaluation, demonstrating consistent, superior performance across diverse datasets and tasks against a well-chosen set of baselines.

However, one reviewer found the motivation for combining the two fields somewhat strained and the overall presentation potentially confusing due to the number of concepts involved. The narrative connecting the high-level motivation to the specific technical solution could be sharpened to be more compelling for all readers. Additionally, the privacy discussion, added during the rebuttal, would be more impactful if it were fully integrated into the paper with a more formal treatment of potential vulnerabilities. Similarly, a quantitative analysis of the computation and communication overhead of the proposed modules would help practitioners assess the method's real-world feasibility.